# Seeing Differently, Acting Similarly: Heterogeneously Observable Imitation Learning

**Xin-Qiang Cai**[1,2]    **Yao-Xiang Ding**[3]    **Zi-Xuan Chen**[1]    **Yuan Jiang**[1] [*]
**Masashi Sugiyama**[4,2]    **Zhi-Hua Zhou**[1]
[1]National Key Laboratory for Novel Software Technology, Nanjing University
[2]The University of Tokyo
[3]State Key Laboratory for CAD&CG, Zhejiang University
[4]RIKEN Center for Advanced Intelligence Project
`cai@ms.k.u-tokyo.ac.jp,dingyx.gm@gmail.com,chenzx@lamda.nju.edu.cn,`
`jiangy@lamda.nju.edu.cn,sugi@k.u-tokyo.ac.jp,zhouzh@lamda.nju.edu.cn`

## Abstract

In many real-world imitation learning tasks, the demonstrator and the learner have to act under *different observation spaces*. This situation brings significant obstacles to existing imitation learning approaches, since most of them learn policies under *homogeneous observation spaces*. On the other hand, previous studies under different observation spaces have strong assumptions that these two observation spaces *coexist during the entire learning process*. However, in reality, the observation coexistence will be limited due to the high cost of acquiring expert observations. In this work, we study this challenging problem with limited observation coexistence under heterogeneous observations: *Heterogeneously Observable Imitation Learning* (HOIL). We identify two underlying issues in HOIL: the dynamics mismatch and the support mismatch, and further propose the *Importance Weighting with REjection* (IWRE) algorithm based on importance weighting and learning with rejection to solve HOIL problems. Experimental results show that IWRE can solve various HOIL tasks, including the challenging tasks of transforming the vision-based demonstrations to random access memory (RAM)-based policies in the Atari domain, even with limited visual observations.

## 1 Introduction

Imitation Learning (IL), which studies how to learn a good policy by imitating the given demonstrations (Xu et al., 2020; Chen et al., 2022), has made significant progress in real-world applications such as autonomous driving (Chen et al., 2019), health care (Iyer et al., 2021), and continuous control (Wang et al., 2023). In tradition, the expert and the learner are assumed to use the same observation space. However, nowadays, many real-world IL tasks demand to remove this assumption (Chen et al., 2019; Warrington et al., 2021), such as in autonomous driving (Chen et al., 2019), recommendation system (Wu et al., 2019), and medical decision making (Wang et al., 2021a). Taking AI for medical diagnosis as an example in Figure 1: A medical AI is learning to make medical decisions based on expert doctor demonstrations. To ensure demonstration quality, the expert may use high-cost observations such as CT, MRI, and B-ultrasound. In contrast, the AI learner is ideal to use only low-cost observations from cheaper devices, which could be newly designed ones that have not been used previously by the expert. Meanwhile, to ensure reliability, it is also reasonable to allow the learner to access the high-cost observations during training under a limited budget (Yu et al., 2019). The above examples share three characteristics: (i) Even though a pair of expert and learner observations can be different, they are under the *same state* of the environment, leading to similar policies; (ii) The learner's new observations are *not available* to the expert when generating demonstrations; (iii) During training, the learner can only access expert observations under a limited budget, in special the high-cost ones, since it is also important to *minimize the usage of the*

---

[*]Corresponding author.

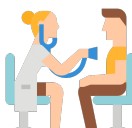 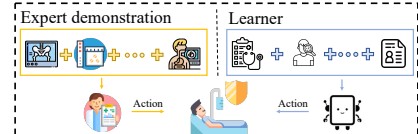

Figure 1: Medical decision making: an example of the HOIL problem. Figures 1, 2, and 3 include some illustrations and pictures from the Internet (source: `https://www.flaticon.com/`).

Table 1: Comparisons between different IL processes. $\mathcal{O}_\mathrm{E}$ and $\mathcal{O}_\mathrm{L}$ denote the observation spaces for experts and learners respectively. N/A denotes not applicable.

| Setting | DSIL | POIL | LBC | HOIL(ours) |
|---|---|---|---|---|
| $\mathcal{O}_\mathrm{E} \neq \mathcal{O}_\mathrm{L}$ | ✗ | ✓ | ✓ | ✓ |
| The demonstrations do not include $\mathcal{O}_\mathrm{L}$ | N/A | ✗ | ✓ | ✓ |
| The learner does not require $\mathcal{O}_\mathrm{E}+\mathcal{O}_\mathrm{L}$ all the time | N/A | ✗ | ✗ | ✓ |
| $\mathcal{O}_\mathrm{E}$ is not more privileged than $\mathcal{O}_\mathrm{L}$ | N/A | ✗ | ✗ | ✓ |

*expert observations* during training to avoid unnecessary costs. We name such IL tasks *Heterogeneously Observable Imitation Learning* (HOIL). Among them, we focus on the most challenging HOIL setting in which *the expert and learner observation spaces have no overlap*.

There are two lines of research studying the related problems, summarized in Table 1 and Figure 2. The first one relates to Domain-shifted IL (DSIL): the observation spaces of experts and learners are the *homogeneous*, while some typical distribution mismatches could exist: morphological, viewpoint, and dynamics mismatch (Stadie et al., 2017; Raychaudhuri et al., 2021). However, the approaches for DSIL are invalid when the observation spaces are *heterogeneous* as in HOIL.

The second line studied IL under different observations similar to HOIL. Some representative works include Partially Observable IL (POIL) (Gangwani et al., 2019; Warrington et al., 2021) and Learning by Cheating (LBC) (Chen et al., 2019). Both POIL and LBC assume that the learner can easily access the expert observations without any budget limit. However, in practice, different from the learner observations, the access to expert observations might be of high cost, invasive, and even unavailable (Yu et al., 2019), which hinder the wide application of these methods.

In this paper, we initialize the study of the HOIL problem. We propose a learning process across observation spaces of experts and learners to solve this problem. We further analyze the underlying issues of HOIL, i.e., the dynamics mismatch and the support mismatch. To tackle both two issues, we use the techniques of *importance weighting* (Shimodaira, 2000; Fang et al., 2020) and *learning with rejection* (Cortes et al., 2016; Geifman & El-Yaniv, 2019) for active querying to propose the *Importance Weighting with REjection* (IWRE) approach. We evaluate the effectiveness of the IWRE algorithm in continuous control tasks of MuJoCo (Todorov et al., 2012), and the challenging tasks of learning random access memory (RAM)-based policies from vision-based expert demonstrations in Atari (Bellemare et al., 2013) games. The results demonstrate that IWRE can significantly outperform existing IL algorithms in HOIL tasks, with limited access to expert observations.

## 2   RELATED WORK

**FESL.** Recently, to deal with the significant challenges in open environment learning (Zhou, 2022), there are emerging studies of feature evolvable stream learning (FESL) (Hou et al., 2017; Hou & Zhou, 2018; Zhang et al., 2020), which are among major inspirations of our work. FESL focuses on supervised learning with heterogeneous feature space. There are also significant differences between existing FESL approaches and HOIL. In (Hou et al., 2017; Hou & Zhou, 2018), they assume that data features are generated from heterogeneous but *fixed and static* data distributions, while in HOIL, the data distributions are *dynamically changing*. FESL with dynamically changing distributions is studied in (Zhang et al., 2020), while they assume that the changes are *passive* to the learner. This is different from HOIL, in which the data distribution changes are *actively* decided by the learner due to the nature of decision-making learning.

**DSIL.** For the standard IL process, where the learner and the expert share the same observation space, current state-of-the-art methods tend to learn the policy in an adversarial style (Wang et al., 2021b;c), like Generative Adversarial Imitation Learning (GAIL) (Ho & Ermon, 2016). When considering the domain mismatch problem, i.e., DSIL, the research aims at addressing the *static distributional shift* of the optimal policies resulted from the environmental differences but still under homogeneous observation spaces. Stadie et al. (2017), Sermanet et al. (2018), and Liu et al. (2018)

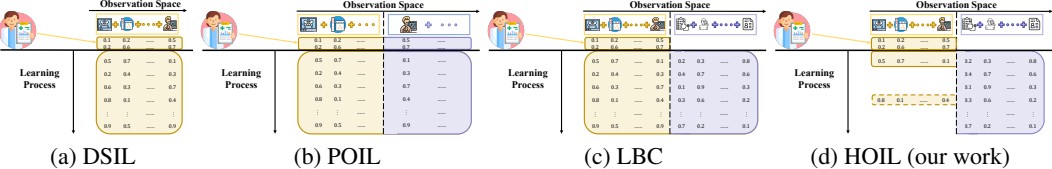

(a) DSIL        (b) POIL        (c) LBC        (d) HOIL (our work)

Figure 2: Comparison sketches of IL with respect to observation spaces. The target of (a) is to learn a policy based on the same observations of the expert, while that of (b), (c), and (d) are to learn a policy based on the second observation space only. The detailed differences can be found in Table 1.

studied the situation where the demonstrations are in view of a third person. Kim et al. (2020) and Kim et al. (2019) addressed the IL problem with morphological mismatch between the expert's and learner's environments. Tirinzoni et al. (2018), Jiang et al. (2020), and Desai et al. (2020) focused on the calibration for the mismatch between simulators and the real world through some transfer learning styles. There are two major differences between HOIL and DSIL: One is that HOIL considers *heterogeneous* observation spaces instead of *homogeneous* ones; another is that without observation heterogeneity, DSIL can directly align two fixed domains, which may not be realistic for solving HOIL when two observation spaces are totally different. Thus HOIL is a significantly more challenging problem than DSIL. Besides, Chen et al. (2019) learned a vision-based agent from a privileged expert. But it can obtain expert observations throughout the whole learning process, so it cannot handle the problem of the support mismatch under HOIL.

**POMDP.** The problem of Partially Observable Markov Decision Process (POMDPs), in which only partial observations are available for the agent(s), has been studied in the context of multi-agent (Omidshafiei et al., 2017; Warrington et al., 2021) and IL (Gangwani et al., 2019; Warrington et al., 2021) problems. But distinct from HOIL, in a POMDP, the learner only has partial observations and shares the *same* underlying observation space with the expert, which would become an obstacle for them to make decisions correctly. For example, Warrington et al. (2021) assumed that the observation of the learner is partial than that of the expert. Instead, in HOIL, expert's and learner's observations are totally *different* from each other, while the learner's observations do not belong to the expert's. For HOIL, the main challenge is to deal with the mismatches between the observation spaces, especially when the access to expert observations is strictly limited.

## 3 THE HOIL PROBLEM

In this section, we first give a formal definition of the HOIL setting, and then introduce the learning process for solving the HOIL problem.

### 3.1 SETTING DEFINITION

A HOIL problem is defined within a Markov decision process with mutiple observation spaces, i.e., $\langle \mathcal{S}, \{\mathcal{O}\}, \mathcal{A}, \mathcal{P}, \gamma \rangle$, where $\mathcal{S}$ denotes the state space, $\{\mathcal{O}\}$ denotes a set of observation spaces, $\mathcal{A}$ denotes the action space, $\mathcal{P} : \mathcal{S} \times \mathcal{A} \times \mathcal{S} \to \mathbb{R}$ denotes the transition probability distribution of the state and action, and $\gamma \in (0, 1]$ denotes the discount factor. Furthermore, a policy $\pi$ over an observation space $\mathcal{O}$ is defined as a function mapping from $\mathcal{O}$ to $\mathcal{A}$, and we denote by $\Pi_{\mathcal{O}}$ the set of all policies over $\mathcal{O}$. In HOIL, both the expert and the learner have their own observation spaces, which are denoted as $\mathcal{O}_{\mathrm{E}}$ and $\mathcal{O}_{\mathrm{L}}$ respectively. Both $\mathcal{O}_{\mathrm{E}}$ and $\mathcal{O}_{\mathrm{L}}$ are assumed to be produced by two bijective mappings $f_{\mathrm{E}} : \mathcal{S} \to \mathcal{O}_{\mathrm{E}}, f_{\mathrm{L}} : \mathcal{S} \to \mathcal{O}_{\mathrm{L}}$, which are unknown functions mapping the underlying true states to the observations. It is obvious to see that by this assumption, any policy over $\mathcal{O}_{\mathrm{E}}$ has a unique correspondence over $\mathcal{O}_{\mathrm{L}}$. This makes HOIL possible since the target of HOIL is to find the policy corresponding to the expert policy under $\mathcal{O}_{\mathrm{L}}$.

A state-action pair $(s, a)$, denoted by $x$, is called an *instance*. Also, a trajectory $\mathcal{T} = \{x_i\}, i := \{1, \ldots, m\}$ is a set of $m$ instances. For each observation space, $\widetilde{x} \in \widetilde{\mathcal{T}} \subseteq \mathcal{O}_{\mathrm{E}} \times \mathcal{A}$ and $\overline{x} \in \overline{\mathcal{T}} \subseteq \mathcal{O}_{\mathrm{L}} \times \mathcal{A}$, where $\mathcal{O}_{\mathrm{E}} = f_{\mathrm{E}}(\mathcal{S})$ and $\mathcal{O}_{\mathrm{L}} = f_{\mathrm{L}}(\mathcal{S})$. Furthermore, we define the *occupancy measure* of a policy $\pi$ under the state space $\mathcal{S}$ as $\rho_\pi : \mathcal{S} \times \mathcal{A} \to \mathbb{R}$ such that $\rho_\pi(x) = \pi(a|o)\mathrm{Pr}(o|s) \sum_{t=0}^{\infty} \gamma^t \mathrm{Pr}(s_t = s|\pi)$. Under HOIL, the learner accesses the expert demonstrations $\widetilde{\mathcal{T}}_{\pi_{\mathrm{E}}}$, a set of instances sampled from $\rho_{\pi_{\mathrm{E}}}$. The goal of HOIL is to learn a policy $\hat{\pi}$ as the corresponding policy of $\pi_{\mathrm{E}}$ over $\mathcal{O}_{\mathrm{L}}$. If $\mathcal{O}_{\mathrm{E}} = \mathcal{O}_{\mathrm{L}}$, HOIL degenerates to standard IL . GAIL (Ho & Ermon, 2016) is one of the state-of-the-art IL approaches under this situation, which tries to minimize the divergence between the learner's

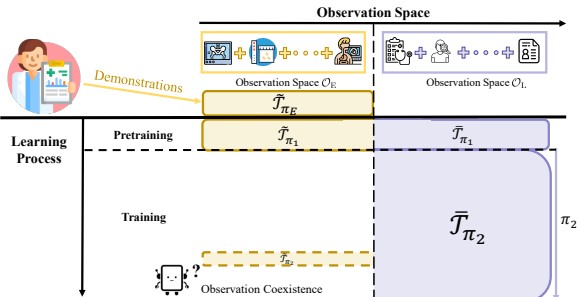

Figure 3: Illustration of a learning process across two different observation spaces for solving HOIL. $\pi_1$ is an auxiliary policy that is additionally provided.

and the expert's occupancy measures $d(\rho_{\hat{\pi}}, \rho_{\pi_\mathrm{E}})$. The objective of GAIL is

$$\min_{\hat{\pi}} \max_w \mathbb{E}_{x \sim \rho_{\pi_\mathrm{E}}}[\log D_w(\widetilde{x})] + \mathbb{E}_{x \sim \rho_{\hat{\pi}}}[\log(1 - D_w(\widetilde{x}))] - \mathbb{H}(\hat{\pi}), \tag{1}$$

where $\mathbb{H}(\hat{\pi})$ is the causal entropy performed as a regularization term, and $D_w : \mathcal{O}_\mathrm{E} \times \mathcal{A} \to [0, 1]$ is the discriminator of $\pi_\mathrm{E}$ and $\hat{\pi}$. GAIL solved Equation (1) by alternatively taking a gradient ascent step to train the discriminator $D_w$, and a minimization step to learn policy $\hat{\pi}$ based on an off-the-shelf reinforcement learning (RL) algorithm with the pseudo reward $-\log D_w(\widetilde{x})$.

## 3.2 THE LEARNING PROCESS FOR SOLVING HOIL

In HOIL, we need to cope with the absence of the learner's observations in demonstrations and the high cost of collecting expert observations while learning. So we introduce a learning process with pretraining across two different observation spaces for solving HOIL, as abstracted in Figure 3.

**Pretraining.** Same as LBC (Chen et al., 2019), we assume that we can obtain an auxiliary policy $\pi_1$ based on $\mathcal{O}_\mathrm{E}$ at the beginning. $\pi_1$ can be directly provided by any sources, or trained by GAIL or behavior cloning (Michie et al., 1990) from online data stream (Cai et al., 2019) or offline demonstrations (Sasaki & Yamashina, 2021) as did in LBC. Besides, we use this $\pi_1$ to sample some data $\mathcal{T}_{\pi_1}$, which contain both observation under $\mathcal{O}_\mathrm{E}$ (i.e., $\widetilde{\mathcal{T}}_{\pi_1}$) and $\mathcal{O}_\mathrm{L}$ (i.e., $\overline{\mathcal{T}}_{\pi_1}$), in order to connect these two different observation spaces. We name $\mathcal{T}_{\pi_1} = \{\widetilde{\mathcal{T}}_{\pi_1}, \overline{\mathcal{T}}_{\pi_1}\}$ the *initial data*.

**Training.** Here we learn a policy $\pi_2$ from the initial data $\overline{\mathcal{T}}_{\pi_1}$ and the collected data $\overline{\mathcal{T}}_{\pi_2}$, under $\mathcal{O}_\mathrm{L}$ only. Besides, the learner is allowed for some operation of *observation coexistence* (OC): At some steps of learning, besides the observations $\mathcal{O}_\mathrm{L}$, the learner could also request $\widetilde{\mathcal{T}}_{\pi_2}$ from the corresponding observations $\mathcal{O}_\mathrm{E}$ (e.g., from the human-understandable sensors). The final objective of HOIL is to learn a good policy $\pi_2$ under $\mathcal{O}_\mathrm{L}$.

In practical applications, the auxiliary policy $\pi_1$ can also come from simulation training or direct imitation. But since $\pi_1$ is additionally provided, it is more practical to consider $\pi_1$ as a non-optimal policy. During training, OC is an essential operation for solving HOIL, which helps the learner address the issues of the dynamics mismatch and the support mismatch (especially the latter one). Also, in reality, we do not need an oracle for actions, which still needs OC for obtaining expert observations first, as in many active querying research (Brantley et al., 2020; Chen et al., 2019), so its cost will be relatively lower.

Besides, the related work (Chen et al., 2019) also required an initialized policy $\pi_1$ to solve their problem, which acts as a teacher under privileged $\mathcal{O}_\mathrm{E}$ in the pretraining and then learned a vision-based student from the guidance of the teacher under both $\mathcal{O}_\mathrm{L}$ and $\mathcal{O}_\mathrm{E}$. Their setting can be viewed as a variety of HOIL with optimal $\pi_1$, unlimited $\mathcal{O}_\mathrm{E}$, and unlimited OC operations, so HOIL is actually a more practical learning framework.

## 4 IMITATION LEARNING WITH IMPORTANCE WEIGHTING AND REJECTION

In HOIL, the access frequency to $\mathcal{O}_\mathrm{E}$ is strictly limited, so it is unrealistic to learn $\pi_2$ in a Dataset Aggregation (DAgger) style (Ross et al., 2011) as in LBC. Therefore, we resort to learning $\pi_2$ with a learned reward function by inverse RL (Abbeel & Ng, 2010) in an adversarial learning style (Ho & Ermon, 2016; Fu et al., 2018).

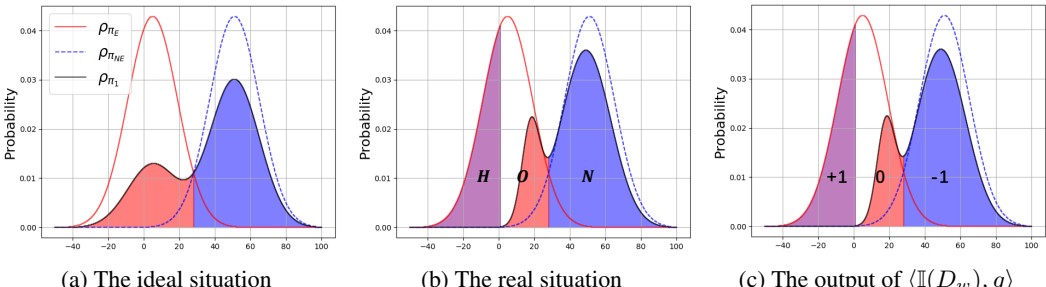

(a) The ideal situation       (b) The real situation       (c) The output of $\langle \mathbb{I}(D_w), g \rangle$

Figure 4: The comparisons among the distributions of expert demonstrations $\rho_{\pi_{\mathrm{E}}}$, initial data $\rho_{\pi_1}$, and non-expert data $\rho_{\pi_{\mathrm{NE}}}$. The red and blue regions denote the expert and non-expert parts of $\rho_{\pi_1}$ respectively. $H$, $O$, and $N$ denote the latent demonstration, the observed demonstration, and the non-expert data respectively. (a) The ideal situation, where $\mathrm{supp}(\rho_{\pi_{\mathrm{E}}}) \setminus \mathrm{supp}(\rho_{\pi_1}) = \varnothing$; (b) The real situation, where $H := \mathrm{supp}(\rho_{\pi_{\mathrm{E}}}) \backslash \mathrm{supp}(\rho_{\pi_1}) \neq \varnothing$ in $\rho_{\pi_{\mathrm{E}}}$; (c) The target output of the combined model $\mathbb{I}[D_w^*]g^*$. The output $+1$, $0$, and $-1$ regions correspond to $H$, $O$, and $N$ respectively.

In addition, both $\mathcal{O}_{\mathrm{E}}$ and $\mathcal{O}_{\mathrm{L}}$ are assumed to share the same latent state space $\mathcal{S}$ as introduced in Section 3.1, so the following analysis will be based on $\mathcal{S}$, while the algorithm will handle the problem based on $\mathcal{O}_{\mathrm{E}}$ and $\mathcal{O}_{\mathrm{L}}$ specifically.

## 4.1 Dynamics Mismatch and Importance Weighting

To analyze the learning process, we let $\rho_{\pi_{\mathrm{E}}}$, $\rho_{\pi_1}$, and $\rho_{\pi_2}$ be the occupancy measure distributions of the expert demonstrations, the initial data, and the data during training respectively. Since we need to consider the sub-optimality of $\pi_1$, $\rho_{\pi_1}$ should be a mixture distribution of the expert $\rho_{\pi_{\mathrm{E}}}$ and non-expert $\rho_{\pi_{\mathrm{NE}}}$, i.e., there exists some $\delta \in (0, 1)$ such that

$$\rho_{\pi_1} = \delta \rho_{\pi_{\mathrm{E}}} + (1 - \delta) \rho_{\pi_{\mathrm{NE}}}, \tag{2}$$

as depicted in Figure 4a. During training, the original objective of $\pi_2$ is to imitate $\pi_{\mathrm{E}}$ through demonstrations. To this end, the original objective of reward function $D_{w_2}$ for $\pi_2$ is to optimize

$$\max_{w_2} \mathbb{E}_{x \sim \rho_{\pi_2}}[\log D_{w_2}(\overline{x})] + \mathbb{E}_{x \sim \rho_{\pi_{\mathrm{E}}}}[\log(1 - D_{w_2}(\overline{x}))]. \tag{3}$$

But the expert demonstrations are only available under $\mathcal{O}_{\mathrm{E}}$. During training, we can only utilize the initial data $\overline{\mathcal{T}}_{\pi_1} \sim \rho_{\pi_1}$ to learn $\pi_2$ and $D_{w_2}$. Besides, as $\pi_1$ is sub-optimal, directly imitating $\overline{\mathcal{T}}_{\pi_1}$ could reduce the performance of the optimal $\pi_2$ to that of $\pi_1$. So we use the importance weighting to calibrate this dynamics mismatch, i.e.,

$$\max_{w_2} \mathcal{L}(D_{w_2}) = \mathbb{E}_{x \sim \rho_{\pi_2}}[\log D_{w_2}(\overline{x})] + \mathbb{E}_{x \sim \rho_{\pi_1}}[\alpha(x) \log(1 - D_{w_2}(\overline{x}))], \tag{4}$$

where $\alpha(x) \triangleq \frac{\rho_{\pi_{\mathrm{E}}}(x)}{\rho_{\pi_1}(x)}$ is an importance weighting factor (Fang et al., 2020). So the current issue lies in how to estimate $\frac{\rho_{\pi_{\mathrm{E}}}}{\rho_{\pi_1}}$ under $\mathcal{O}_{\mathrm{E}}$. To achieve this purpose, we need to bridge the expert demonstrations and the initial data. Therefore, here we use these two data sets to train an adversarial model $D_{w_1}$ in the same way as $D_{w_2}$ in the pretraining:

$$\max_{w_1} \mathcal{L}(D_{w_1}) \triangleq \mathbb{E}_{x \sim \rho_{\pi_1}}[\log D_{w_1}(\widetilde{x})] + \mathbb{E}_{x \sim \rho_{\pi_{\mathrm{E}}}}[\log(1 - D_{w_1}(\widetilde{x}))]. \tag{5}$$

If we write the training criterion (5) in the form of integral, i.e.,

$$\max_{w_1} \mathcal{L}(D_{w_1}) = \int_x [\rho_{\pi_1} \log D_{w_1} + \rho_{\pi_{\mathrm{E}}} \log(1 - D_{w_1})] dx, \tag{6}$$

then, by setting the derivative of the objective to 0 ($\frac{\partial \mathcal{L}}{\partial D_{w_1}} = 0$), we can obtain the optimum $D_{w_1}$:

$$D_{w_1}^* = \frac{\rho_{\pi_1}}{\rho_{\pi_1} + \rho_{\pi_{\mathrm{E}}}}, \tag{7}$$

in which the order of differentiation and integration was changed by the Leibniz rule. Besides, we can sufficiently train $D_{w_1}$ using the initial data $\widetilde{\mathcal{T}}_{\pi_1}$ and the expert demonstrations $\widetilde{\mathcal{T}}_{\pi_E}$. Then $D_{w_1}$ will be good enough to estimate the importance weighting factor, i.e.,

$$\alpha(x) \triangleq \frac{\rho_{\pi_E}}{\rho_{\pi_1}} = \frac{1 - D_{w_1}^*(\widetilde{x})}{D_{w_1}^*(\widetilde{x})} \approx \frac{1 - D_{w_1}(\widetilde{x})}{D_{w_1}(\widetilde{x})}. \tag{8}$$

In this way, we can use $D_{w_1}$, which can connect demonstrations and initial data, to calibrate the learning process of $D_{w_2}$. The final optimization objective for $D_{w_2}$ is

$$\max_{w_2} \mathcal{L}(D_{w_2}) = \mathbb{E}_{x \sim \rho_{\pi_2}} \log D_{w_2}(\overline{x}) + \mathbb{E}_{x \sim \rho_{\pi_1}} \frac{1 - D_{w_1}(\widetilde{x})}{D_{w_1}(\widetilde{x})} \log[1 - D_{w_2}(\overline{x})]. \tag{9}$$

In this way, $D_{w_2}$ can effectively dig out the expert part of $\rho_{\pi_1}$ and produce efficient rewards for $\pi_2$.

### 4.2 Support Mismatch

So far, the challenges have still been similar to homogeneously observable IL. However, our preliminary experiments demonstrated that mere importance weighting is not enough to fix the problem that occurred by the absence of interactions under $\mathcal{O}_E$. So there exist some other issues between the expert demonstrations and the initial data. To find out the underlying issues, we plotted the t-distributed Stochastic Neighbor Embedding (t-SNE) (van der Maaten & Hinton, 2008) visualizations of these two empirical distributions under $\mathcal{O}_E$ on *Hopper* and *Walker2d* in Figure 5. Twenty trajectories were collected for both the expert demonstrations and the initial data. We can observe that there exist some high-density regions of demonstrations in which the initial data do not cover; that is, there exist some regions of the demonstrations that $\pi_1$ did *not explore*. Wang et al. (2019) found a similar phenomenon in the standard IL setting. On the other hand, the importance weighting $\alpha$ cannot calibrate this situation where $\frac{\rho_{\pi_E}}{\rho_{\pi_1}} = \infty$.

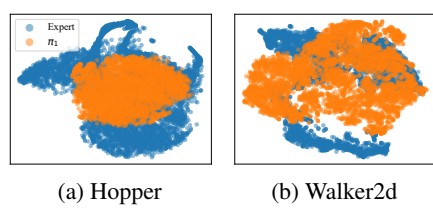

(a) Hopper    (b) Walker2d

Figure 5: t-SNE visualizations of expert demonstrations and collected data of $\pi_1$ under $\mathcal{O}_E$.

To formulate this problem, here we introduce the *support set* of the occupancy measure:

**Definition 1** (Support Set). *The support set of an occupancy measure $\rho$ is the subset of the domain containing the elements which are not mapped to zero:*

$$\text{supp}(\rho) := \{x \in \mathcal{S} \times \mathcal{A} | \rho(x) \neq 0\}. \tag{10}$$

Due to the sub-optimality of $\pi_1$, $\text{supp}(\rho_{\pi_E}) \setminus \text{supp}(\rho_{\pi_1}) \neq \varnothing$ (see Figure 4b). We call this part the *latent demonstration*, defined as:

**Definition 2** (Latent Demonstration). *The latent demonstration $H$ is the set of the domain that belongs to the relative complement of $\text{supp}(\rho_{\pi_1})$ in $\text{supp}(\rho_{\pi_E})$:*

$$H := \{x \in \mathcal{S} \times \mathcal{A} | \text{supp}(\rho_{\pi_E}) \setminus \text{supp}(\rho_{\pi_1})\}. \tag{11}$$

Also, another part of the demonstration is named the *observed demonstration*, defined as:

**Definition 3** (Observed Demonstration). *The observed demonstration $O$ is the set of the domain that belongs to the complement of $H$ in $\text{supp}(\rho_{\pi_E})$:*

$$O := \{x \in \mathcal{S} \times \mathcal{A} | \text{supp}(\rho_{\pi_E}) \cap \text{supp}(\rho_{\pi_1})\}. \tag{12}$$

Besides, the data outside of demonstrations should be non-expert data:

**Definition 4** (Non-Expert Data). *The non-expert data $N$ is the set of the domain out of $\text{supp}(\rho_{\pi_E})$:*

$$N := \{x \in \mathcal{S} \times \mathcal{A} | \rho_{\pi_E}(x) = 0\}. \tag{13}$$

In other words, the sub-optimality of $\pi_1$ will cause not only the dynamics mismatch, but also the appearance of the latent demonstration $H$. We call the latter one the problem of *support mismatch*. Intuitively, *when $\pi_2 \to \pi_E$, we have $H \to \varnothing$, monotonously*. So in order to fix the support mismatch between $\rho_{\pi_E}$ and $\rho_{\pi_1}$, *guiding $\pi_2$ to find out $H$ is the key*.

In addition, the support mismatch problem can be viewed as an inverse problem of the out-of-distribution (OOD) problem that frequently occurred in offline RL setting (Levine et al., 2020), in which they tried to avoid $\text{supp}(\rho_{\pi_1}) \setminus \text{supp}(\rho_{\pi_E})$ instead.

## 4.3 IMITATION LEARNING WITH REJECTION

We can observe that $H \cup O \cup N = \mathcal{S} \times \mathcal{A}$. So it is desirable to filter out $H$ from $O$ and $N$. Meanwhile, $D_{w_1}$ and $D_{w_2}$ can only classify $O \cup H$ and $N$, under $\mathcal{O}_E$ and $\mathcal{O}_L$ respectively. Therefore, here we design two models $g_1 : \mathcal{O}_E \times \mathcal{A} \to \{0, 1\}$ and $g_2 : \mathcal{O}_L \times \mathcal{A} \to \{0, 1\}$ (Output 0: $x \in O$ and output 1: otherwise), so that given $x \sim \mathcal{T}$ (corresponding $\widetilde{x} \sim \widetilde{\mathcal{T}}$ and $\overline{x} \sim \overline{\mathcal{T}}$), they can satisfy

$$H = \{x \in \mathcal{S} \times \mathcal{A} | \mathbb{I}[D_{w_1}^*(\widetilde{x})]g_1^*(\widetilde{x}) = \mathbb{I}[D_{w_2}^*(\overline{x})]g_2^*(\overline{x}) = +1\}, \tag{14}$$

$$O = \{x \in \mathcal{S} \times \mathcal{A} | \mathbb{I}[D_{w_1}^*(\widetilde{x})]g_1^*(\widetilde{x}) = \mathbb{I}[D_{w_2}^*(\overline{x})]g_2^*(\overline{x}) = 0\}, \tag{15}$$

$$N = \{x \in \mathcal{S} \times \mathcal{A} | \mathbb{I}[D_{w_1}^*(\widetilde{x})]g_1^*(\widetilde{x}) = \mathbb{I}[D_{w_2}^*(\overline{x})]g_2^*(\overline{x}) = -1\}, \tag{16}$$

where $\mathbb{I}[\cdot]$ takes $+1$ if $\cdot > 0.5$, and $-1$ otherwise. $\mathbb{I}[D_w^*(x)]g^*(x)$ is depicted in Figure 4c.

To this end, both $g_1$ and $g_2$ should be able to cover $O$, meanwhile $g_2$ can be adaptive to the continuous change of $\rho_{\pi_2}$ due to the update of $\pi_2$. Here we learn $g_1$ and $g_2$ in a rejection form, to *reject $O$ from $O \cup H$* (where $\mathbb{I}(D_w) = +1$). Concretely, the rejection setting is the same as in Cortes et al. (2016). Also, inspired by Geifman & El-Yaniv (2019), the optimization objective of $D_w$ and $g$ is

$$\mathcal{L}(D_w, g) \triangleq \hat{l}(D_w, g) + \lambda \max(0, c - \hat{\phi}(g))^2, \tag{17}$$

where $c > 0$ denotes the target coverage, and $\lambda$ denotes the factor for controlling the relative importance of rejection. Besides, the empirical coverage $\hat{\phi}(g)$ is defined as

$$\hat{\phi}(g|X) \triangleq \frac{1}{m} \sum_{i=1}^{m} g(x_i), \tag{18}$$

with a batch of data $X = \{x_i\}, i \in [m]$. The empirical rejection risk $\hat{l}(D_w, g)$ is the ratio between the covered risk of the discriminator and the empirical coverage:

$$\hat{l}(D_w, g) \triangleq \frac{\frac{1}{m} \sum_{i=1}^{m} \langle \mathcal{L}(D_w(x_i)), g(x_i) \rangle}{\hat{\phi}(g)}. \tag{19}$$

Meanwhile, both $D_{w_1}$ and $g_1$ can access $\rho_{\pi_E}$ under $\mathcal{O}_E$ directly. So given $\overline{x} \sim \overline{\mathcal{T}}_{\pi_2}$ under $\mathcal{O}_L$, once $\langle \mathbb{I}(D_{w_2}(\overline{x})), g_2(\overline{x}) \rangle = +1$, we can query the corresponding observations $\widetilde{x}$ of $\overline{x}$ through OC operation and use $\langle \mathbb{I}(D_{w_1}(\widetilde{x})), g_1(\widetilde{x}) \rangle$ to calibrate the output of $g_2$ and $D_{w_2}$. In this way, $g_2$ and $D_{w_2}$ can be entangled together and adaptively guide $\pi_2$ to find out the latent demonstrations $H$ under $\mathcal{O}_L$.

## 4.4 IWRE

Here we combine importance weighting and rejection into a unified procedure, resulting in a novel algorithm named Importance Weighting with REjection (IWRE). Concretely, in a HOIL process:

**Pretraining.** We train a discriminator $D_{w_1}$ by Equation (5) and its corresponding rejection model $g_1$ by Equation (17) using the initial data and the expert demonstrations.

**Training.** We train a discriminator $D_{w_2}$ by the combination of Equation (9) and Equation (17), as well as its corresponding rejection model $g_2$ by Equation (17), using the initial data, the data collected by $\pi_2$, and the output of $D_{w_1}$ with $g_1$ through OC operation. Also, $\pi_2$ will be updated with $D_{w_2}$ and $g_2$ asymmetrically as in GAIL.

The pseudo-code of our algorithm is provided in the appendix.

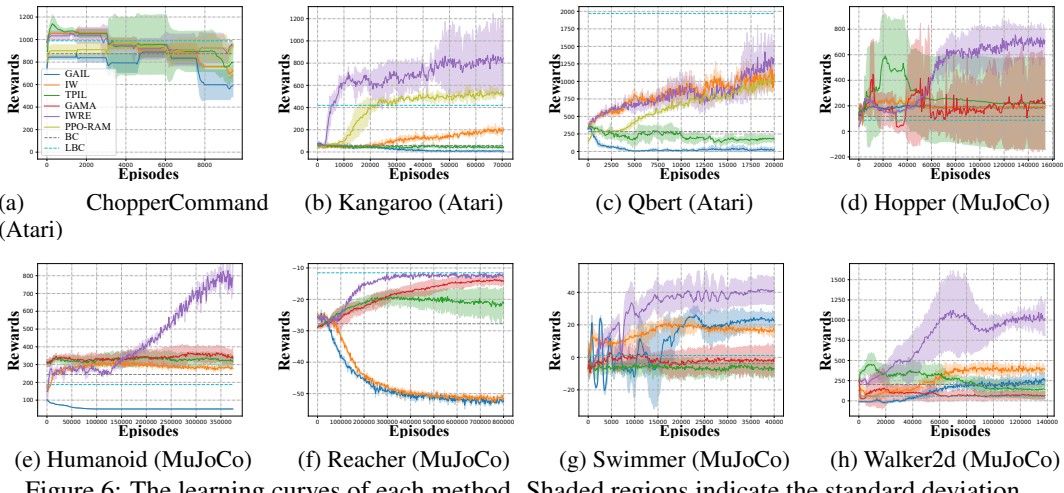

| (a) | ChopperCommand (Atari) | (b) Kangaroo (Atari) | (c) Qbert (Atari) | (d) Hopper (MuJoCo) |
| (e) Humanoid (MuJoCo) | (f) Reacher (MuJoCo) | (g) Swimmer (MuJoCo) | (h) Walker2d (MuJoCo) |

Figure 6: The learning curves of each method. Shaded regions indicate the standard deviation.

## 5 EXPERIMENT

In this section, we validate our algorithm in Atari 2600 (Bellemare et al., 2013) (GPL License) and MuJoCo (Todorov et al., 2012) (Academic License) environments. The experiments were designed to investigate: 1) Can IWRE achieve significant performance under HOIL tasks? 2) Can IWRE deal with the support mismatch problem? 3) During training, is active querying for HOIL indeed necessary? Below we first introduce the experimental setup and then investigate the above questions. More results and experimental details are included in the appendix.

### 5.1 EXPERIMENTAL SETUP

**Environments.** We choose three pixel-memory based games in Atari and five continuous control objects in MuJoCo on OpenAI platform (Brockman et al., 2016) (MIT License). For pixel-memory Atari games, $\mathcal{O}_E$: $84 \times 84 \times 4$ raw pixels; $\mathcal{O}_L$: 128-byte random access memories (RAM). For continuous control MuJoCo objects, $\mathcal{O}_E$: half of original observation features; $\mathcal{O}_L$: another half of original observation features. Besides, twenty expert trajectories were collected for each environment. Each result contains five trials with different random seeds. All experiments were conducted on server clusters with NVIDIA Tesla V100 GPUs. Further details of the environments are included in the appendix.

**Baselines.** Six basic contenders were included in the experiments: Vanilla **GAIL** (Ho & Ermon, 2016), GAIL with importance weighting (Fang et al., 2020) (**IW**, an ablation study of IWRE), third-person IL (Stadie et al., 2017) (**TPIL**), generative adversarial MDP alignment (Kim et al., 2020) (**GAMA**), behavioral cloning (Bain & Sammut, 1996) (**BC**), and learning by cheating (Chen et al., 2019) (**LBC**). For IW, we utilized $D_{w_1}$ trained in the pretraining to calculate the importance weight; also the optimization objective for $D_{w_2}$ during training is the same as Equation (9); TPIL learns the third-person demonstrations by leading the cross-entropy loss into the update of the feature extractor; GAMA learns a mapping function $\psi$ in view of adversarial training to align the observation of the target domain into the source domain, and thereby can utilize the policy in the source domain for zero-shot imitation. For fairness, we allowed the interaction between the policy and the environment for GAMA under HOIL; LBC uses $\pi_1$ learned from privileged states as a teacher to train $\pi_2$ in a DAgger (Ross et al., 2011) style, so here we allowed LBC to access $\mathcal{O}_E$ during the whole IL process. In Atari, to investigate whether our method could achieve good performance for RAM-based control, we further included a contender **PPO-RAM**, which uses proximal policy optimization (PPO) (Schulman et al., 2017) to perform RL directly with environmental true rewards under the RAM-based observations. More detailed setup including query strategies for TPIL and GAMA, network architecture, and hyper-parameters are reported in the appendix.

### 5.2 RESULTS

Experimental results are reported in Figure 6. Since the mapping function is hard to learn when input is RAM and output is raw images, we omit the results of GAMA in Atari. We can observe

that while IW is better than GAIL in most environments, both GAIL and IW can hardly outperform $\pi_1$. Because they just imitated the performance of $\pi_1$ instead of $\pi_E$, even with importance weighting for calibration. For TPIL, its learning process was extremely unstable on *Hopper*, *Swimmer*, and *Walker2d* due to the continuous distribution shift. Furthermore, the performance of GAMA was not satisfactory in *Hopper* and *Walker2d* because its mapping function is hard to learn well when the support mismatch appears. The results of TPIL and GAMA demonstrate that DSIL methods will be invalid under heterogeneous observations as in HOIL tasks. On Atari environments, $\mathcal{O}_E$ contains more privileged information than $\mathcal{O}_L$, so LBC can achieve good performance. But when $\mathcal{O}_E$ is not more privileged than $\mathcal{O}_L$, like in most environments of MuJoCo, its performance will decrease due to the support mismatch, which would make it even worse than BC. Finally, IWRE obtained the best performance on 6/8 environments, and comparable performance with LBC on *Reacher*, which shows the effectiveness of our method even with limited access to $\mathcal{O}_E$ (LBC can access to $\mathcal{O}_E$ all the time). Besides, we can see that the performance differences between the GAIL/IW and IWRE/TPIL/GAMA/LBC are huge (especially on *Reacher*) because of the absence of queries, which demonstrates that the query operation is indeed necessary for HOIL problems.

Moreover, even learned with true rewards, PPO-RAM surprisingly failed to achieve comparable performance to IWRE, which shows that IWRE could possibly learn more effective rewards than true environmental rewards in RAM-input tasks. The results verify that, IWRE provides a powerful approach for tackling HOIL problems, even under the situation that the demonstrations are gathered from such a different observation space, meanwhile $\mathcal{O}_E$ is strictly limited during training.

**t-SNE visualization of $\rho_{\pi_2}$ and $\rho_{\pi_E}$ under $\mathcal{O}_E$.** In Section 4.2, we point that the sub-optimality of $\pi_1$ will cause the problem of support mismatch, which is embodied as the appearance of the latent demonstration $H$ during training. Also the empirical results in Figure 5 on *Hopper* and *Walker2d* verify the existence of this problem. So we want to investigate whether the superiority of IWRE indeed comes from successfully tackling the support mismatch problem. To this end, we plotted the t-SNE visualization of the same expert demonstrations as in Section 4.2 and the collected data of $\pi_2$ by IWRE under $\mathcal{O}_E$ ($\mathcal{O}_E$ is hidden to $\pi_2$). All setups are the same as in Section 4.2. From the results shown in Figure 7, we can see that even under $\mathcal{O}_E$, which cannot be obtained by $\pi_2$, al-

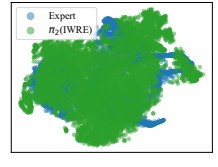 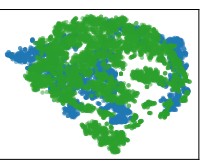

(a) Hopper  (b) Walker2d

Figure 7: t-SNE visualizations of expert demonstrations and collected data of $\pi_2$ under $\mathcal{O}_E$. The high-density regions of the expert demonstrations were covered by the collected data of $\pi_2$ of IWRE.

most all high-density regions of the demonstrations were covered by the collected data. Meanwhile, the latent demonstration $H$ is dug out nearly. The results demonstrate that IWRE basically solves the problem of support mismatch and thereby performs well in these environments.

Besides, some collected data of $\pi_2$ of IWRE were out of the distribution of the demonstrations, which means $\pi_2$ slightly overly explored the environment. Since $\mathcal{O}_E$ is hidden to $\pi_2$, the reward function will encourage $\pi_2$ to explore more areas to fix the support mismatch problem. Meanwhile, the out-of-distribution problem in HOIL is not as severe as in the offline RL settings (Levine et al., 2020), so this over-exploration phenomenon makes sense.

## 6   CONCLUSION

In this paper, we proposed a new learning framework named *Heterogeneously Observable Imitation Learning* (HOIL), to formulate situations where the observation space of demonstrations differs from that of the imitator while learning. We formally modeled the learning process of HOIL, in which access to expert observations is limited due to the high cost. Furthermore, we analyzed the underlying challenges of HOIL: the dynamics mismatch and the support mismatch, on the occupancy distributions between the demonstrations and the policy. To tackle these challenges, we proposed a new algorithm named Importance Weighting with REjection (IWRE), using importance weighting and learning with rejection. Experimental results showed that the direct imitation and domain adaptive methods could not solve this problem, while our approach obtained promising results. In the future, we hope to give the theoretical guarantee for our algorithm IWRE and investigate how many $\mathcal{O}_E$ we need to query to learn a promising $\pi_2$. Furthermore, we hope to use the learning framework of HOIL and IWRE to tackle more learning scenarios with demonstrations in different spaces.

ACKNOWLEDGMENT

This research was supported by: National Key Research and Development Program of China (2020AAA0109401); National Science Foundation of China (62176117, 61921006, 62206245); the Institute for AI and Beyond, UTokyo; JST SPRING, Grant Number JPMJSP2108. The authors would like to thank Guoqing Liu, Yoshihiro Nagano, and the anonymous reviewers for their insightful comments and suggestions.

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

## A NOTATIONS

The notations of the main paper are gathered in Table 2.

Table 2: The notations of the main paper.

| Notation | Meaning |
|---|---|
| $\mathcal{S}$ | State space |
| $\mathcal{A}$ | Action space |
| $\mathcal{O}$ | Observation space |
| $\mathcal{O}_{\mathrm{E}}$ | Observation space of the expert's view |
| $\mathcal{O}_{\mathrm{L}}$ | Observation space of the learner's view |
| $\mathcal{P}$ | Transition Probability |
| $\gamma$ | Discounted factor |
| $\pi_{\mathrm{E}}$ | Expert policy under $\mathcal{O}_{\mathrm{E}}$ |
| $\pi_1$ | Auxiliary policy under $\mathcal{O}_{\mathrm{E}}$ |
| $\pi_2$ | Target policy under $\mathcal{O}_{\mathrm{L}}$ |
| $f_{\mathrm{E}}$ | Mapping function $\mathcal{S} \rightarrow \mathcal{O}_{\mathrm{E}}$ |
| $f_{\mathrm{L}}$ | Mapping function $\mathcal{S} \rightarrow \mathcal{O}_{\mathrm{L}}$ |
| $\widetilde{\mathcal{T}}_{\pi_{\mathrm{E}}}$ | Trajectory sampled by $\pi_{\mathrm{E}}$ under $\mathcal{O}_{\mathrm{E}}$ (demonstrations) |
| $\widetilde{\mathcal{T}}_{\pi_1}$ | Trajectory sampled by $\pi_1$ under $\mathcal{O}_{\mathrm{E}}$ |
| $\widetilde{\mathcal{T}}_{\pi_2}$ | Trajectory sampled by $\pi_2$ under $\mathcal{O}_{\mathrm{E}}$ |
| $\overline{\mathcal{T}}_{\pi_1}$ | Trajectory sampled by $\pi_1$ under $\mathcal{O}_{\mathrm{L}}$ |
| $\overline{\mathcal{T}}_{\pi_2}$ | Trajectory sampled by $\pi_2$ under $\mathcal{O}_{\mathrm{L}}$ |
| $x$ | An instance of state-action pair |
| $\widetilde{x}$ | An instance of observation-action pair under $\mathcal{O}_{\mathrm{E}}$ |
| $\overline{x}$ | An instance of observation-action pair under $\mathcal{O}_{\mathrm{L}}$ |
| $\rho_{\pi_{\mathrm{E}}}$ | Occupancy measure of the expert policy $\pi_{\mathrm{E}}$ |
| $\rho_{\pi_1}$ | Occupancy measure of the auxiliary policy $\pi_1$ |
| $\rho_{\pi_2}$ | Occupancy measure of the target policy $\pi_2$ |
| $D_{w_1}$ | Adversarial model on $\widetilde{\mathcal{T}}_{\pi_{\mathrm{E}}}$ and $\widetilde{\mathcal{T}}_{\pi_1}$ |
| $D_{w_2}$ | Adversarial model on $\overline{\mathcal{T}}_{\pi_1}$ and $\overline{\mathcal{T}}_{\pi_2}$ |
| $\alpha$ | Importance weighting factor |
| $H$ | Latent demonstration |
| $O$ | Observed demonstration |
| $N$ | Non-expert data |
| $g_1$ | rejection model under $\mathcal{O}_{\mathrm{E}}$ |
| $g_2$ | rejection model under $\mathcal{O}_{\mathrm{L}}$ |

# B  ALGORITHM

The pseudo codes of our algorithm are illustrated in Algorithms 1 and 2.

---

**Algorithm 1** IWRE.Pretraining

---

**Input:** Auxiliary policy $\pi_1$; Expert demonstrations $\widetilde{\mathcal{T}}_{\pi_{\mathrm{E}}}$.
**Output:** Evolving data $\{\widetilde{\mathcal{T}}_{\pi_1}, \overline{\mathcal{T}}_{\pi_1}\}$; Discriminator $D_{w_1}$; Rejection model $g_1$.
 1: **function** IWRE.PRETRAINING($\pi_1$)
 2:     Sample the evolving data $\{\widetilde{\mathcal{T}}_{\pi_1}, \overline{\mathcal{T}}_{\pi_1}\} \sim \rho_{\pi_1}$ by $\pi_1$.
 3:     Train $D_{w_1}$ and $g_1$ by Equation (5) and (17) respectively using $\widetilde{\mathcal{T}}_{\pi_{\mathrm{E}}}$ and $\widetilde{\mathcal{T}}_{\pi_1}$.
 4:     **return** $\overline{\mathcal{T}}_{\pi_1}, D_{w_1}, g_1$
 5: **end function**

---

**Algorithm 2** IWRE.Training

---

**Input:** Expert demonstrations $\widetilde{\mathcal{T}}_{\pi_{\mathrm{E}}}$; Evolving data $\overline{\mathcal{T}}_{\pi_1}$; Discriminator $D_{w_1}$; Rejection model $g_1$.
**Output:** Target policy $\pi_2$.
 1: **function** IWRE.TRAINING($\widetilde{\mathcal{T}}_{\pi_{\mathrm{E}}}, \overline{\mathcal{T}}_{\pi_1}, D_{w_1}, g_1$)
 2:     Initialize $\pi_2$, $D_{w_2}$, and $g_2$.
 3:     **for** each step $t$ **do**
 4:         Sample $\overline{\mathcal{T}}_{\pi_2} \sim \rho_{\pi_2}$ by $\pi_2$.
 5:         **for** each mini-batch $\{\overline{x}_{\pi_2}\}$ and $\{\overline{x}_{\pi_1}\}$ from $\overline{\mathcal{T}}_{\pi_2}$ and $\overline{\mathcal{T}}_{\pi_1}$ **do**
 6:             Update $\pi_2$ by RL algorithms (such as PPO (Schulman et al., 2017)) using instances $\{\overline{x}_{\pi_2}\}$ and pseudo rewards $\{-\log D_{w_2}(\overline{x}_{\pi_2})\}$.
 7:             Update $D_{w_2}$ by Equation (9) using negative instances $\{\overline{x}_{\pi_2}\}$ and positive ones $\{\overline{x}_{\pi_1}\}$.
 8:             **if** $\langle \mathbb{I}(D_{w_2}(\overline{x}_{\pi_2})), g_2(\overline{x}_{\pi_2}) \rangle = +1$ **then**
 9:                 Query the $\mathcal{O}_{\mathrm{E}}$ observation of $\overline{x}_{\pi_2}$, i.e., $\widetilde{x}_{\pi_2}$, through OC operation.
10:                 Update $D_{w_2}$ and $g_2$ by Equation (17) using the instance $\overline{x}_{\pi_2}$ and the corresponding label $\langle \mathbb{I}(D_{w_1}(\widetilde{x}_{\pi_2})), g_1(\widetilde{x}_{\pi_2}) \rangle$.
11:             **end if**
12:         **end for**
13:     **end for**
14:     **return** $\pi_2$
15: **end function**

---

# C  DEFINITIONS

The core challenges of HOIL, i.e., dynamics mismatch and support mismatch, are illustrated as below.

**Definition 5** (Dynamics Mismatch). *The dynamics mismatch between the demonstrations and the initial data denotes the situation that:*

$$\frac{\rho_{\pi_{\mathrm{E}}}}{\rho_{\pi_1}} = \frac{\pi_{\mathrm{E}}(a|o) \sum_{t=0}^{\infty} \gamma^t \mathrm{Pr}(s_t = s|\pi_{\mathrm{E}})}{\pi_1(a|o) \sum_{t=0}^{\infty} \gamma^t \mathrm{Pr}(s_t = s|\pi_1)} \neq 1. \tag{20}$$

**Definition 6** (Support Mismatch). *The support mismatch between the demonstrations and the initial data denotes the situation that:*

$$\mathrm{supp}(\rho_{\pi_{\mathrm{E}}}) \setminus \mathrm{supp}(\rho_{\pi_1}) = \{x \in \mathcal{S} \times \mathcal{A} | \rho_{\pi_{\mathrm{E}}}(x) \neq 0\} \setminus \{x \in \mathcal{S} \times \mathcal{A} | \rho_{\pi_1}(x) \neq 0\} \neq \varnothing. \tag{21}$$

Table 3: Environmental summary of the tasks.

| Environment | Observation Space $\mathcal{O}_E$ | Observation Space $\mathcal{O}_L$ | Expert Rewards |
|---|---|---|---|
| Qbert | | | $4750.00 \pm 50.51$ |
| ChopperCommand | $84 \times 84 \times 4$(image) | $128$(unsigned int) | $3135.00 \pm 145.86$ |
| Kangaroo | | | $4175.00 \pm 94.21$ |
| Hopper | 8 | 9 | $709.96 \pm 75.54$ |
| Humanoid | 4 | 4 | $539.20 \pm 26.26$ |
| Reacher | 5 | 6 | $-8.99 \pm 0.54$ |
| Swimmer | 5 | 6 | $52.24 \pm 1.29$ |
| Walker2d | 188 | 188 | $929.97 \pm 24.09$ |

Table 4: Comparisons between all contenders and IWRE in HOIL.

| Algorithm | Considering heterogeneous observations | Being able to query | Not requiring $\mathcal{O}_E$ all along |
|---|---|---|---|
| GAIL | ✗ | ✗ | ✓ |
| GAIL-Rand | ✗ | ✓ | ✓ |
| IW | ✗ | ✗ | ✓ |
| IW-Rand | ✗ | ✓ | ✓ |
| TPIL | ✗ | ✓ | ✓ |
| GAMA | ✗ | ✓ | ✓ |
| BC | ✗ | ✗ | ✓ |
| LBC | ✓ | ✓ | ✗ |
| PPO-RAM | ✗ | ✗ | ✓ |
| IWRE | ✓ | ✓ | ✓ |

## D  DETAILED SETUP FOR THE EXPERIMENTS

**Environment and Contenders.** For the environments that used in the main body of the paper:

1. **Pixel-memory Atari games**. $\mathcal{O}_E$: $84 \times 84 \times 4$ raw pixels; $\mathcal{O}_L$: 128-byte random access memories (RAM). Expert: converged DQN-based agents (Mnih et al., 2013). Atari games contain two totally isolated views: raw pixels and RAM, under the same state. Through these environments, we want to investigate whether the agent can learn an effective policy from demonstrations under completely different observation spaces. Moreover, IL with visual observations only is already very difficult (Cai et al., 2021), while learning a RAM-based policy can be even more challenging (Bellemare et al., 2013; Sygnowski & Michalewski, 2016), so few IL research reported desirable results on this task.

2. **Continuous control MuJoCo objects**. $\mathcal{O}_E$: half of original observation features; $\mathcal{O}_L$: another half of original observation features. Expert: converged DDPG-based agents (Lillicrap et al., 2016). The features of MuJoCo contain monotonous information like the direction, position, velocity, etc., of an object. Here we want to investigate whether the agent can learn from demonstrations with complementary signals under observations with missing information. Meanwhile, we make sure RL algorithms can obtain comparable performances under $\mathcal{O}_E$ and $\mathcal{O}_L$.

The details of the environments are reported in Table 3. Also, the detailed comparisons of the contenders (both in the main paper and the appendix) and IWRE are gathered in Table 4.

**Learning process.** To simulate the situation that $\mathcal{O}_E$ is costly, the steps for training $\pi_1$ was set as 1/4 of that for training $\pi_2$, using GAIL (Ho & Ermon, 2016)/HashReward (Cai et al., 2021) under the $\mathcal{O}_E$ space for MuJoCo/Atari environments. The learning steps were $10^7$ for MuJoCo and $5 \times 10^6$ for Atari environments. In the pretraining, we sampled 20 trajectories from $\pi_1$, and the data from each trajectory had both $\mathcal{O}_E$ and $\mathcal{O}_L$ observations. In the training, each method learned $4 \times 10^7$ steps for MuJoCo and $2 \times 10^7$ steps for Atari under the $\mathcal{O}_L$ space to obtain $\pi_2$.

**Query Strategy.** For TPIL and GAMA, if the output of the domain invariant discriminator is larger than 0.5, which means the encoder fails to generate proper features to confuse its discriminator, then we would query $\mathcal{O}_E$ of this data to update the encoder. For IWRE, the threshold of the rejection model $g$ and the discriminator $D_{w_2}$ was also 0.5, which means that if $g_2(\overline{x}) > 0.5$ meanwhile $D_{w_2}(\overline{x}) > 0.5$, $\mathcal{O}_E$ of this data would be queried. $D_{w_2}$, $\pi_2$, and the encoder (for TPIL/GAMA)

were pretrained for 100 epochs for all methods using evolving data during pretraining. The basic RL algorithm is PPO, and the reward signals of all methods were normalized into $[0, 1]$ to enhance the performance of RL (Dhariwal et al., 2017). The buffer size for TPIL and IWRE was set as 5000. Each time the buffer is full, the encoder and the rejection model will be updated for 4 epochs; also LBC will update $\pi_2$ for 100 epochs with batch size 256 using the cross-entropy loss for Atari and the mean-square loss for MuJoCo. We set all hyper-parameters, update frequency, and network architectures of the policy part the same as Dhariwal et al. (2017). Besides, the hyper-parameters of the discriminator for all methods were the same: The rejection model and discriminator were updated using Adam with a decayed learning rate of $3 \times 10^{-4}$; the batch size was 256. The ratio of update frequency between the learner and discriminator was 3: 1. The target coverage $c$ in Equation (17) was set as $0.8$. $\lambda$ in Equation (17) was 1.0.

## E  RL PERFORMANCE UNDER THE DIVISIONS OF MUJOCO

Here we report the performance under the division of $\mathcal{O}_E$ and $\mathcal{O}_L$ in MuJoCo. The details of the division are reported in Table 5. We use DDPG-based (Lillicrap et al., 2016) agent with $10^7$ training steps and repeat 10 times with different random seeds. The results are shown in Figure 8. We can see that the agent can obtain comparable performance under $\mathcal{O}_E$ and $\mathcal{O}_L$. So for MuJoCo environments, the fairness of the division in HOIL can be guaranteed, and $\mathcal{O}_E$ is not more or less privileged than $\mathcal{O}_L$.

Table 5: The observation division into $\mathcal{O}_E$ and $\mathcal{O}_L$ in MuJoCo. The numbers denote the randomly selected observation indexes in the corresponding MuJoCo environment on OpenAI Gym (Brockman et al., 2016) platform.

| | $\mathcal{O}_E$ | $\mathcal{O}_L$ |
|---|---|---|
| Walker2d | [5, 7, 8, 10, 11, 14, 15, 16] | [0, 1, 2, 3, 4, 6, 9, 12, 13] |
| Swimmer | [0, 3, 6, 7] | [1, 2, 4, 5] |
| Reacher | [0, 1, 7, 8, 10] | [2, 3, 4, 5, 6, 9] |
| Hopper | [1, 3, 6, 7, 9, 10] | [0, 2, 4, 5, 8] |
| Humanoid | [2, 3, 5, 6, 7, 10, 11, 12, 13, 16, 18, 19, 22, 23, 25, 29, 31, 32, 34, 36, 37, 40, 43, 44, 45, 47, 48, 49, 51, 54, 56, 57, 61, 63, 65, 66, 67, 68, 77, 78, 82, 86, 87, 89, 90, 93, 94, 95, 97, 98, 99, 102, 103, 108, 110, 112, 113, 117, 119, 120, 121, 122, 123, 124, 126, 127, 128, 133, 135, 144, 146, 147, 148, 151, 152, 153, 158, 160, 161, 162, 166, 167, 170, 171, 173, 174, 176, 177, 178, 180, 181, 184, 185, 187, 188, 191, 194, 198, 199, 200, 201, 202, 207, 208, 209, 210, 211, 212, 214, 215, 219, 223, 227, 228, 229, 231, 232, 233, 234, 236, 237, 238, 242, 244, 246, 248, 251, 253, 257, 258, 259, 260, 262, 264, 265, 267, 268, 271, 272, 273, 275, 278, 279, 280, 281, 285, 287, 289, 290, 291, 293, 294, 296, 299, 302, 304, 305, 306, 307, 308, 311, 312, 313, 315, 316, 319, 322, 326, 328, 329, 332, 337, 342, 343, 344, 345, 349, 358, 361, 362, 364, 365, 366, 368, 370, 372, 373, 375] | [0, 1, 4, 8, 9, 14, 15, 17, 20, 21, 24, 26, 27, 28, 30, 33, 35, 38, 39, 41, 42, 46, 50, 52, 53, 55, 58, 59, 60, 62, 64, 69, 70, 71, 72, 73, 74, 75, 76, 79, 80, 81, 83, 84, 85, 88, 91, 92, 96, 100, 101, 104, 105, 106, 107, 109, 111, 114, 115, 116, 118, 125, 129, 130, 131, 132, 134, 136, 137, 138, 139, 140, 141, 142, 143, 145, 149, 150, 154, 155, 156, 157, 159, 163, 164, 165, 168, 169, 172, 175, 179, 182, 183, 186, 189, 190, 192, 193, 195, 196, 197, 203, 204, 205, 206, 213, 216, 217, 218, 220, 221, 222, 224, 225, 226, 230, 235, 239, 240, 241, 243, 245, 247, 249, 250, 252, 254, 255, 256, 261, 263, 266, 269, 270, 274, 276, 277, 282, 283, 284, 286, 288, 292, 295, 297, 298, 300, 301, 303, 309, 310, 314, 317, 318, 320, 321, 323, 324, 325, 327, 330, 331, 333, 334, 335, 336, 338, 339, 340, 341, 346, 347, 348, 350, 351, 352, 353, 354, 355, 356, 357, 359, 360, 363, 367, 369, 371, 374] |

## F  ESTIMATION OF $H$, $O$, AND $N$ BY $\mathbb{I}[D_{w_2}]g_2$

To investigate the ability of IWRE to distinguish the areas of latent demonstrations $H$, observed demonstrations $O$, and non-expert data $N$ during policy learning, we recorded the accuracy and estimated ratio of each part on *Hopper* and *Walker2d*. The calculations of each curve are shown as below:

$$\text{Accuracy\_}H = \frac{\sum_{i=1}^{m}\{\mathbb{I}[D_{w_1}(\widetilde{x}_i)]g_1(\widetilde{x}_i) == 1 \&\& \mathbb{I}[D_{w_2}(\overline{x}_i)]g_2(\overline{x}_i) == 1\}}{\sum_{i=1}^{m}\{\mathbb{I}[D_{w_1}(\widetilde{x}_i)]g_1(\widetilde{x}_i) == 1\}}, \qquad (22)$$

$$\text{Accuracy\_}O = \frac{\sum_{i=1}^{m}\{\mathbb{I}[D_{w_1}(\widetilde{x}_i)]g_1(\widetilde{x}_i) == 0 \&\& \mathbb{I}[D_{w_2}(\overline{x}_i)]g_2(\overline{x}_i) == 0\}}{\sum_{i=1}^{m}\{\mathbb{I}[D_{w_1}(\widetilde{x}_i)]g_1(\widetilde{x}_i) == 0\}}, \qquad (23)$$

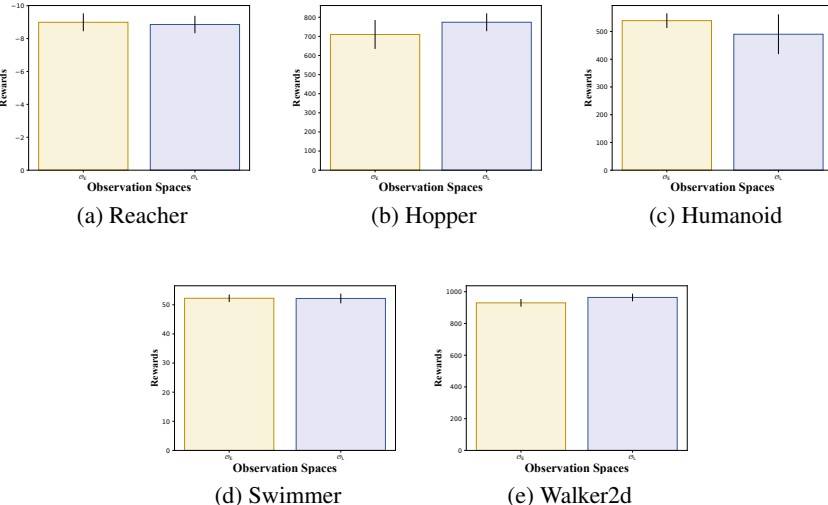

(a) Reacher         (b) Hopper         (c) Humanoid

(d) Swimmer         (e) Walker2d

Figure 8: The performance of RL methods under the division of $\mathcal{O}_\mathrm{E}$ and $\mathcal{O}_\mathrm{L}$ in MuJoCo. The agent can obtain comparable performances under $\mathcal{O}_\mathrm{E}$ and $\mathcal{O}_\mathrm{L}$, so that we can make sure the fairness of the experiment of HOIL in the main paper.

$$\text{Accuracy\_}N = \frac{\sum_{i=1}^{m}\{\mathbb{I}[D_{w_1}(\widetilde{x}_i)]g_1(\widetilde{x}_i) == -1 \&\& \mathbb{I}[D_{w_2}(\overline{x}_i)]g_2(\overline{x}_i) == -1\}}{\sum_{i=1}^{m}\{\mathbb{I}[D_{w_1}(\widetilde{x}_i)]g_1(\widetilde{x}_i) == -1\}}, \tag{24}$$

$$\text{Ratio\_}H = \frac{\sum_{i=1}^{m}\{\mathbb{I}[D_{w_1}(\widetilde{x}_i)]g_1(\widetilde{x}_i) == 1\}}{m}, \tag{25}$$

$$\text{Ratio\_}O = \frac{\sum_{i=1}^{m}\{\mathbb{I}[D_{w_1}(\widetilde{x}_i)]g_1(\widetilde{x}_i) == 0\}}{m}, \tag{26}$$

$$\text{Ratio\_}N = \frac{\sum_{i=1}^{m}\{\mathbb{I}[D_{w_1}(\widetilde{x}_i)]g_1(\widetilde{x}_i) == -1\}}{m}, \tag{27}$$

in which $\{\overline{x}_i, \widetilde{x}_i\} \sim \rho_{\pi_2}$ denotes a batch of data sampled by $\pi_2$. The results are shown in Figure 9. The results depicted not only the accuracies of $\mathbb{I}[D_{w_2}]g_2$, but also the changes of these three areas during the policy learning. We can see that the accuracies in each area and the ratio of $O$ will decrease at first. While at the same time, the ratio of $H$ will increase. This is because the successful detection of $H$ will decrease the estimated ratio of $O$ and reduce the accuracy of $\mathbb{I}[D_{w_2}]g_2$. With the help of query operations, the accuracy of $\mathbb{I}[D_{w_2}]g_2$ will gradually increase. Also, followed by the learning procedure of the policy $\pi_2$, more and more $H$ will be recognized as $O$, with less and less $N$. This is why in the following period, the ratios of $H$ and $N$ will decrease while that of $O$ will increase. These results also verify that our algorithm IWRE can indeed detect $H$, $O$, and $N$ successfully as the learning process of the policy.

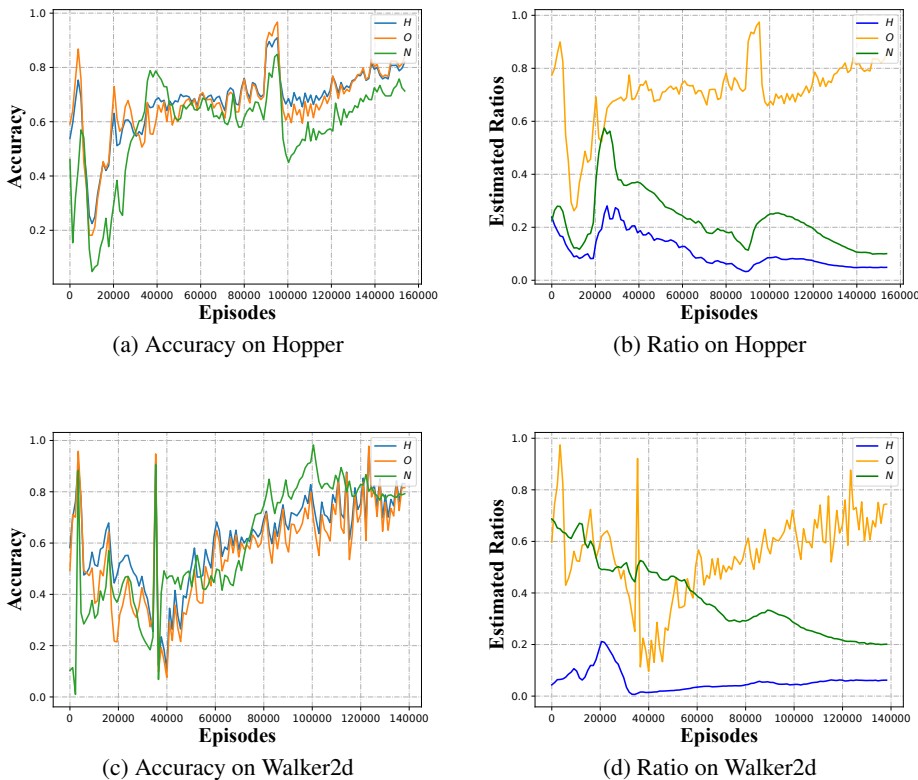

Figure 9: The accuracies and ratios of $H$, $O$, and $N$ calculated by $\mathbb{I}[D_{w_1}(\widetilde{x}_i)]g_1(\widetilde{x}_i)$ and $\mathbb{I}[D_{w_2}(\overline{x}_i)]g_2(\overline{x}_i)$ during policy learning.

## G INVESTIGATIONS ON *ChopperCommand* PERFORMANCE

We observed some interesting phenomenons on *ChopperCommand* performance. Here we investigate the behavior of a random policy, the IWRE policy, and the expert policy, shown in Figure 10. On *ChopperCommand*, the agent needs to imitate the expert to manipulate a helicopter (noted by the black circle) to avoid attacks from enemy aircraft meanwhile shooting them. So the policy in this environment can be split into two parts:

1. The movement of the helicopter;
2. The shoot of the helicopter.

While capturing the image-level semantic information with only RAM input is quite difficult. Even with RAM input, IWRE still successfully learned the expert's movement of the helicopter (see the rows of "IWRE 2e7 steps" and "Expert Policy" in Figure 10). But it is much harder to capture the shot bullet in the image, not to mention in the RAM. So IWRE policy did not shoot enemy aircraft successfully. However, the environmental reward of *ChopperCommand* is only related to the number of enemy aircraft shot down, regardless of the distance the helicopter has flown. Therefore, in Figure 6a, IWRE policy can obtain higher rewards than the random policy at the beginning of the training (1e5 steps). Then the environmental reward fluctuates within a specific range. On the other hand, IWRE indeed learned some expert's policy of manipulating the helicopter in view of the trajectory shown in Figure 10. So we believe that only considering the reward value as an evaluation is insufficient to reflect the degree of imitation on *ChopperCommand*. This interesting phenomenon also reveals that the reward function learned from IL can be very different from the environmental reward functions in RL tasks, since the IL reward can capture expert behaviors that are not reflected by the environmental reward signals. Figure 10 can also be used as a metric. We also provide the

full videos of these four policies on *ChopperCommand* in this anonymous link: `https://www.dropbox.com/sh/fz7xpbt4y8t3umz/AACw5cMq5eG8sCl04x9_OOnha?dl=0`.

Figure 10: The sequence comparisons between a random policy, the IWRE policy with 1e5/2e7 steps, and the expert policy. The agent needs to manipulate a helicopter (noted by the black circle) to avoid attacks from enemy aircraft meanwhile shooting them. The timestamp for each column of images and seed for each environment is the same.

## H    QUERY EFFICIENCY

We also investigate whether our query strategy is efficient. To this end, we allocate the query budget, i.e., limiting the query ratio for each method. For TPIL, it preferentially queries those data with low $D_{w_\phi}$ output; for our method IWRE, it preferentially queries those data with high $\langle D_{w_2}, g_2 \rangle$ output. Besides, since GAIL and IW cannot directly perform queries, we design a random-selection strategy for them as GAIL-Rand and IW-Rand: for each batch of data, we randomly select data and input the $\mathcal{O}_E$ observations of these data to $D_{w_1}$. If $D_{w_1}(\overline{x}) > 0.5$, which means $D_{w_1}$ regard this data being belonging to the expert demonstrations, then we would label this data as the expert data to update $D_{w_2}$. The results are depicted in Figure 11.

We can observe that the random strategy does not always improve the performance of GAIL and IW. For GAIL-Rand, without importance weighting to calibrate the learning process of the reward function, its performance become even worse on *Hopper*, *Swimmer*, and *Walker2d*, because the queried information enhances the discrimination ability of reward function, making it even more impossible for the agent to obtain effective feedbacks; for IW-Rand, its performance is better than GAIL-Rand on most environments, and is reinforced on *Hopper*, *Reacher*, and *Walker2d*, which further demonstrate that the query operation is indeed necessary for HOIL problem, but still fails compared with our method; for TPIL, it is comparable with IW-Rand, however, its performance improvement is very limited as the budget increases, and on *Swimmer* and *Walker2d* there even exist performance degradations, which suggests that its query strategy is very unstable; for GAMA, it has a good start point, but the performance gain is very limited while the budget increases; for our method, its performance is almost the same as that of IW-Rand without query on most environments. When it is allowed to query $\mathcal{O}_E$ observation, our method outperforms other methods with a large gap, which shows that the query strategy of our method is indeed more efficient.

## I    IMITATION WITH DIFFERENT NUMBER OF EXPERT TRAJECTORIES

The performances of different numbers of expert trajectories of all contenders are reported in Figure 12. Each experiment is conducted 5 trials with different random seeds. We can observe that even with a very limited number of trajectories, our algorithm achieves better performance than other algorithms in most environments.

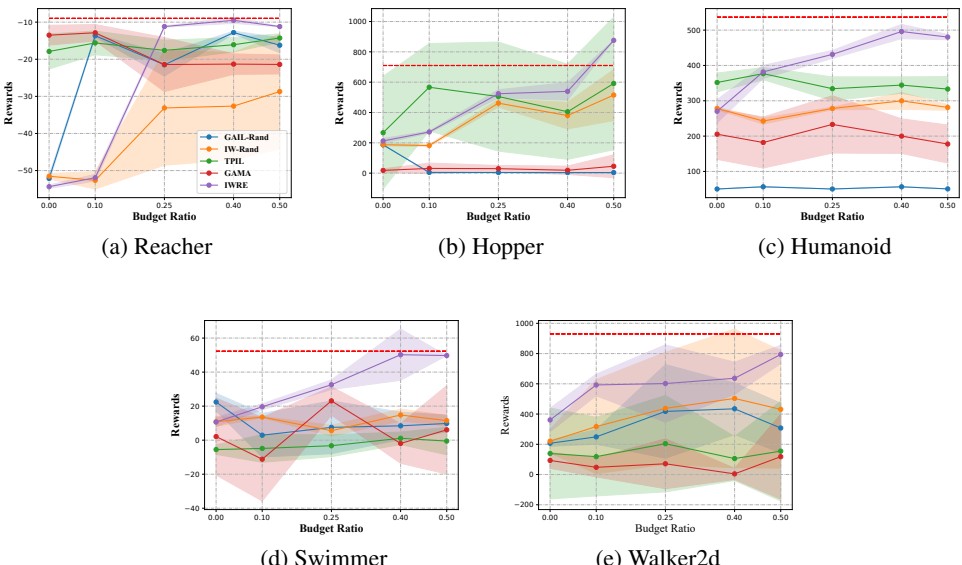

Figure 11: The final rewards of each method on MuJoCo with different budget ratios, where the shaded regions indicate the standard deviation. The red horizontal dotted line represents the averaged performance of the expert.

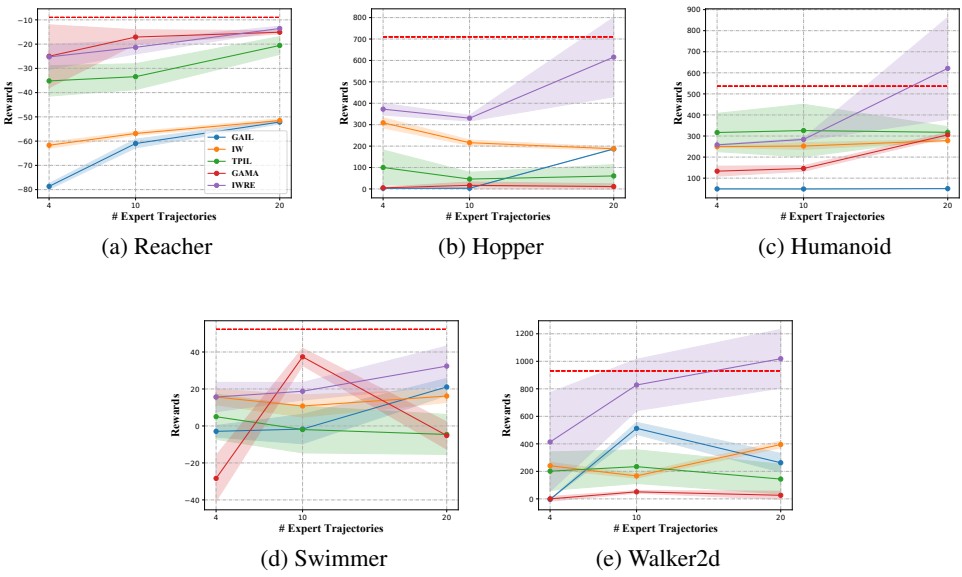

Figure 12: The learning curves of each method in MuJoCo environments with different numbers of expert trajectories, where the shaded region indicates the standard deviation.

