# OpenReview forum: "Seeing Differently, Acting Similarly: Heterogeneously Observable Imitation Learning"
_ICLR.cc/2023/Conference — ICLR 2023 notable top 25%_

### Official Review · Reviewer_ntXD · 2022-10-25

**Confidence:** 2
**Correctness:** 4
**Technical Novelty And Significance:** 3
**Empirical Novelty And Significance:** 3
**Recommendation:** 6

**Clarity, Quality, Novelty And Reproducibility:**

The paper has good clarity.

The proposed algorithm is novel.

The method is reproducible.


**Strength And Weaknesses:**

Strength:

The proposed method is explained with details. Both dynamics and support set mismatch are handled with the proposed novel algorithm. Experiments show evidence for the improved covering rate in expert data by the learned policy. Explanations for the experimental results are thorough.


Weakness:

For the t-SNE plot in Fig. 5, is it true that the support set of $\pi_1$ also has some part outside the expert data? How is this part handled by the method? This does not seem to be captured by Fig. 4.

Adding ablation studies with only importance weighting or the rejection will help to understand their effects better. However, no such ablation study is conducted in the current paper.

The fonts in the figures are too small to read.

In Fig. 6 (a), the ChopperCommand results are a bit weird. Is it too easy or too hard to learn? It seems the reward becomes large very quickly but decreases along the training. Why is this happening? Please explain more in the paragraph.


**Summary Of The Paper:**

The paper proposes a method for imitation learning with heterogeneous observations. Specifically, the dynamics mismatch and the support mismatch are handled using importance weighting and rejection. Experiments show the effectiveness of the proposed IWRE method on both Atari and MuJoCo tasks.

**Summary Of The Review:**

The paper proposes an effective and novel algorithm for handling both dynamics and support set mismatch in imitation learning. The experiments justify the claims that the proposed method helps with a better coverage of the expert data as well as the effectiveness in facing the heterogeneous observations. Some questions need to be answered well in the paragraph. Adding ablation studies will make it a better work.

---

> ### Author Response · Authors · 2022-11-15
> **Authors Response**
>
> We thank Reviewer ntXD for the comments and suggestions. We have uploaded the revision that incorporated your suggestions. The updates are highlighted in blue. Below we would like to respond to the technology issues.
>
> **Q1**:
>
> >For the t-SNE plot in Fig. 5, is it true that the support set of $\pi_1$ also has some part outside the expert data? How is this part handled by the method? This does not seem to be captured by Fig. 4.
>
> **A1**:
>
> Thank you for your detailed discussion! The outside part of $\pi_1$ data is actually the non-expert part, corresponding to the "N" part in Fig. 4. Through our algorithm, the importance weight of this part is $\alpha=\frac{\rho_{\pi_1}}{\rho_{\pi_{\mathrm{E}}}}=0$, so that the policy can avoid learning this part as the expert demonstrations.
>
> **Q2**:
>
> >Adding ablation studies with only importance weighting or the rejection will help to understand their effects better. However, no such ablation study is conducted in the current paper.
>
> **A2**:
>
> We totally agree that the ablation study is necessary, as follows:
>
> 1. The ablation study with only the importance weighting. Actually, we have included this ablation of our algorithm as **IW** in the experiment. Also, the experimental phenomenon is consistent with the statement of the paper, i.e., without the calibration of the rejection model, the performance of IW is only a little better than that of GAIL.
> 2. The ablation study with only the rejection model. For IWRE, the rejection model plays the role of enhancing the importance weighting performance, so we did not include the rejection model as an independent module for ablation study.
>
> **Q3**:
>
> >In Fig. 6 (a), the ChopperCommand results are a bit weird. Is it too easy or too hard to learn? It seems the reward becomes large very quickly but decreases along the training. Why is this happening? Please explain more in the paragraph.
>
> **A3**:
>
> Thank you for your detailed discussions on the experiments. We did further investigations on the ChopperCommand experiment and found some interesting but reasonable phenomena. The investigations are also gathered in Appendix G of the revision.
>
> Here we investigate the behavior of a random policy, the IWRE policy, and the expert policy, shown in the figure of an anonymous link: https://www.dropbox.com/sh/fz7xpbt4y8t3umz/AACw5cMq5eG8sCl04x9_OOnha?dl=0. On ChopperCommand, the agent needs to imitate the expert to manipulate a helicopter (noted by the black circle) to avoid attacks from enemy aircraft meanwhile shooting them. So the policy in this environment can be split into two parts:
>
> 1. The movement of the helicopter;
> 2. The shoot of the helicopter.
>
> While capturing the image-level semantic information with only RAM input is quite difficult. Even with RAM input, IWRE still successfully learned the expert's movement of the helicopter (see the rows of "IWRE 2e7 steps" and "Expert Policy" in the figure). But it is much harder to capture the shot bullet in the image, not to mention in the RAM. So IWRE policy did not shoot enemy aircraft successfully. However, the environmental reward of ChopperCommand is only related to the number of enemy aircraft shot down, regardless of the distance the helicopter has flown. Therefore, in Fig. 6 (a), IWRE policy can obtain higher rewards than the random policy at the beginning of the training (1e5 steps). Then the reward fluctuates within a specific range. On the other hand, IWRE indeed learned some expert's policy of manipulating the helicopter in view of the trajectory shown in the figure. So we believe that only considering the reward value as an evaluation is insufficient to reflect the degree of imitation on ChopperCommand. This interesting phenomenon also reveals that the reward function learned from IL can be very different from the environmental reward functions in RL tasks, since the IL reward can capture expert behaviors that are not reflected by the environmental reward signals. The figure in the link can also be used as a metric. We also provide the full videos of these four policies on ChopperCommand in the same anonymous link: https://www.dropbox.com/sh/fz7xpbt4y8t3umz/AACw5cMq5eG8sCl04x9_OOnha?dl=0.

---

> > ### Author Response · Authors · 2022-11-30
> > **Looking forward to further discussions**
> >
> > We thank Reviewer ntXD for the reviews and comments. Based on our responses and further experiments, we would like to ask if we have addressed Reviewer ntXD's concerns.

---

### Official Review · Reviewer_j2UZ · 2022-10-26

**Confidence:** 4
**Correctness:** 3
**Technical Novelty And Significance:** 3
**Empirical Novelty And Significance:** 1
**Recommendation:** 3

**Clarity, Quality, Novelty And Reproducibility:**

Clarity:
- The writing is decent, however I think the paper would be easier to understand if pseudocode for the full algorithm were presented in the main paper rather than the appendix.

Quality/Clarity:
- As mentioned above, the algorithm is quite complicated - I believe a simpler algorithm would be preferable and easier to build upon. Overall the experiments seem sound given the assumptions made by this work.

Novelty:
- To my knowledge, this combination of importance weighting and rejection sampling is novel.

Reproducibility:
- The authors do not mention any code release in their reproducibility statement.

**Strength And Weaknesses:**

# Strengths:
- The goal of learning from demonstrations gathered in a different observation space is definitely interesting
- To my knowledge, this combination of importance weighting and rejection sampling is novel.


# Weaknesses:
- I'm not convinced by the setup they're considering here. They make assumptions which seem unrealistic to me: in particular, they assume it is possible to interact with *both* the expert and agent MDPs to some extent. This is necessary for gathering what they call the evolving dataset, which is a set of trajectories where the observations in both MDPs are aligned. I find this to be a strong assumption, and one which other methods (e.g. [1]) do not make - they assume expert trajectories are given and only interaction with the learner's MDP is allowed.

For this reason, although I think the general goal of IL from heterogeneous observations is very interesting (for example, learning from YouTube videos where the perspective of the expert is different from that of the agent), this paper makes overly strong assumptions that one can gather trajectories from both MDPs. This strong assumption limits the significance and impact of this work. Also, the fact that they are only considering discriminator-based approaches and not others (e.g. non-adversarial ones such as RED [2] or DRIL [3]) limits the applicability.

Also, the algorithm proposed is fairly complicated: there are two different learning phases with GAIL subroutines as well as training a classifier on intermediate data. I believe the community would be more open to adopting a simpler, more end-to-end approach.

[1] https://arxiv.org/pdf/1703.01703.pdf
[2] http://proceedings.mlr.press/v97/wang19d/wang19d.pdf
[3] https://openreview.net/pdf?id=rkgbYyHtwB

**Summary Of The Paper:**

This paper proposes a method for imitation learning (IL) when the expert and agent operate from different observation spaces. The paper proposes a new algorithm for dealing with this setting, which relies on two components: an importance weighting correction which modifies the standard GAIL objective, and a rejection mechanism to address potential support mismatch between expert and policy distributions. The algorithm is evaluated on 5 Mujoco and 3 Atari environments.

The algorithm operates in two stages. First, GAIL is run in the expert's MDP to learn a decent, but not necessarily optimal policy. Then, the same policy is run in the expert and learner MDPs (this is assumed to be possible), giving a dataset of aligned trajectories. This is then used to compute the importance weighting correction factor. Finally, a classifier is trained to filter out the non-expert parts of the policy, which is used for the rejection sampling mechanism.

**Summary Of The Review:**

Overall, I do not recommend acceptance for this paper due to:
- the overly strong assumptions made by this work, which somewhat undermine the otherwise interesting setting they consider
- the complexity of the algorithm

---

> ### Author Response · Authors · 2022-11-15
> **Authors Response**
>
> We thank Reviewer j2UZ for the comments and suggestions. But some comments are mismatched about our paper. Below we would like to clarify these mismatches.
>
> **Q1**:
>
> >They make assumptions which seem unrealistic to me: in particular, they assume it is possible to interact with both the expert and agent MDPs to some extent. This is necessary for gathering what they call the evolving dataset, which is a set of trajectories where the observations in both MDPs are aligned. I find this to be a strong assumption, and one which other methods(e.g. [1]) do not make - they assume expert trajectories are given and only interaction with the learner's MDP is allowed.
>
> **A1**:
>
> Such an assumption does not appear in our paper. In our setting, only one MDP, instead of two, exists. Also, the situation that the agent can access two different observations under the same state is commonly considered in other related works ([1, 2]), while we limit the number of accesses to the expert observations. On the other hand, we also compared the method proposed in [1] (TPIL) in our experiment, which failed to solve the HOIL problem. We have also compared our setting with other researches clearly in Table 1. So our setting actually considers a weaker situation than other works.
>
> **Q2**:
>
> >The algorithm operates in two stages. First, GAIL is run in the expert's MDP to learn a decent, but not necessarily optimal policy. Then, the same policy is run in the expert and learner MDPs (this is assumed to be possible), giving a dataset of aligned trajectories.
>
> and
>
> >Also, the algorithm proposed is fairly complicated: there are two different learning phases with GAIL subroutines as well as training a classifier on intermediate data.
>
> **A2**:
>
> We note that the description of the algorithm did not match ours. In our algorithm, the auxiliary policy can be provided in any form. In the pretraining phase, we use this auxiliary policy to sample some data that contain both the expert and the learner observations in one single MDP (since they are under the same states, which is also discussed in **A1**). Then, we use the sampled data to train other models and the target policy. In summary, our algorithm follows the "pretrain then fine-tune" paradigm, which is commonly used in the machine learning community.
>
> [1] Third-Person Imitation Learning. Stadie et al. ICLR 2017.
>
> [2] Learning by Cheating. Chen et al. CoRL 2019.

---

> > ### Author Response · Authors · 2022-11-30
> > **Looking forward to further discussions**
> >
> > We thank Reviewer j2UZ for the reviews and comments. Based on our clarifications, we would like to ask if there are other concerns that make Reviewer j2UZ still feel reject.

---

> > > ### Comment · Reviewer_j2UZ · 2022-11-30
> > > **Thanks for the response**
> > >
> > > Thanks for the response. However, my concerns still remain, especially the first.
> > >
> > > Yes, I agree there is one MDP but two observation spaces. The assumption I find unrealistic is stated here, at the bottom of page 3:
> > >
> > > "Besides, we use this $\pi_1$ to sample some data $\mathcal{T}_{\pi_1}$ , which contain both observation under $\mathcal{O}_E$ and $\mathcal{O}_L$ in order to connect these two different observation spaces."
> > >
> > > This is a very strong assumption. One of the main appeals of being able to perform IL from mismatched viewpoints is to learn from large amounts of observational data from uncurated datasets, which will likely have diverse viewpoints (for example, youtube videos, traffic videos, etc). However, assuming that we can execute the _same_ policy in both viewpoints severely limits these possibilities.
> > >
> > > Furthermore, I am pretty sure that [1] does _not_ make this assumption. If I am mistaken, please point out the exact part of that paper where they state that assumption.
> > >
> > > Thank you for clarifying that the auxiliary policy can be provided beforehand. I think the paper would be a lot clearer if the algorithm pseudocode in Appendix B would be in the main paper.

---

> > > > ### Author Response · Authors · 2022-12-01
> > > > **Clarifications on the misunderstandings of the assumption**
> > > >
> > > > Thanks for the follow-up comments and agreement about our clarifications on previous reviews. Below we are glad to further clarify some misunderstanding points about the assumption. For the convenience of explanation, we copy Table 1 in the main paper here, which compares related works.
> > > >
> > > > Table 1: Comparisons between different IL processes. $\mathcal{O}_\mathrm{E}$ and $\mathcal{O}_\mathrm{L}$ denote the observation spaces for experts and learners respectively.
> > > > |                                                                                             |         |            |            |            |
> > > > |:-------------------------------------------------------------------------------------------:|:-------:|:----------:|:----------:|:----------:|
> > > > | Setting                                                                                     | DSIL    | POIL       | LBC        | HOIL(ours) |
> > > > | $\mathcal{O}_\mathrm{E} \neq \mathcal{O}_\mathrm{L}$                                        | &#10006; | &#10004; | &#10004; | &#10004; |
> > > > | The demonstrations do not include $\mathcal{O}_\mathrm{L}$                                  | &#10006; | &#10006;    | &#10004; | &#10004; |
> > > > | The learner does not require $\mathcal{O}_\mathrm{E}$+$\mathcal{O}_\mathrm{L}$ all the time | &#10006; | &#10006;    | &#10006;    | &#10004; |
> > > > | $\mathcal{O}_\mathrm{E}$ is not more privileged than $\mathcal{O}_\mathrm{L}$               | &#10006; | &#10006;    | &#10006;    | &#10004; |
> > > > |                                                                                             |         |            |            |            |
> > > >
> > > > 1.
> > > >
> > > > > Strong assumption
> > > >
> > > > This assumption is not strong but necessary in heterogeneous observation settings. The comment “execute the same policy in both viewpoints” corresponds to sampling data with both expert and learner observations using the auxiliary policy in our paper. We note that many related researches with successful applications, such as LBC [2] and POIL [3] [4], also need data with both expert and learner observations. Our work takes a step further and considers a more challenging setting, as shown in Table 1. These works, including ours, build upon this assumption because many real-world applications naturally satisfy the assumption, such as autonomous driving [2], recommendation system [5], and medical decision making [6].
> > > >
> > > > 2.
> > > >
> > > > > [1] does not make this assumption
> > > >
> > > > [1] belongs to Domain-Shifted IL (DSIL). As shown in Table 1, [1] does not consider that expert and learner observations have different feature spaces as in [2], [3], [4], and our work. Since the learning scenario is different, the other assumptions in [1] are not comparable to that in our work, as well as [2][3][4]. The detailed difference can be found both in our previous response (A1 in “Authors Response” to Reviewer j2UZ) and the main paper (Sections 1 and 2, as well as the experiment of [1] in Section 5).
> > > >
> > > > [1] Third-Person Imitation Learning. Stadie et al. ICLR 2017.
> > > >
> > > > [2] Learning by Cheating. Chen et al. CoRL 2019.
> > > >
> > > > [3] Learning belief representations for imitation learning in pomdps. Gangwani et al., UAI 2019.
> > > >
> > > > [4] Robust asymmetric learning in pomdps. ICML 2021.
> > > >
> > > > [5] A recommender system for heterogeneous and time sensitive environment. Wu et al. RecSys 2019.
> > > >
> > > > [6] A hierarchical fusion framework to integrate homogeneous and heterogeneous classifiers for medical decision-making. Wang et al. Knowl. Based Syst 2021.

---

> > > > > ### Author Response · Authors · 2022-12-08
> > > > > **Kindly request feedback**
> > > > >
> > > > > We thank Reviewer j2UZ for the abundant discussions. Overall, the points of our previous response are as follows:
> > > > >
> > > > > 1. It is a not strong but common assumption in the heterogeneous observation learning problems;
> > > > >
> > > > > 2. [1] considered domain-shifted IL setting, which is different from heterogeneous observations in the research line that our work belongs to.
> > > > >
> > > > > Again thanks for your comments. We will make it more clear in the revision. Since it remains less than one week for the discussion, we would like to gently request your feedback and discussions if you have other concerns that make you feel reject still.
> > > > >
> > > > > [1] Third-Person Imitation Learning. Stadie et al. ICLR 2017.

---

> > > > > > ### Comment · Reviewer_j2UZ · 2022-12-09
> > > > > > **Table 1 seems wrong**
> > > > > >
> > > > > > Thanks for the response.
> > > > > >
> > > > > > Table 1 seems wrong to me. Let's go through it one row at a time.
> > > > > >
> > > > > > - Row 1: Table 1 says that for [1], $\mathcal{O}_E \neq \mathcal{O}_L$ does not hold - in other words, $\mathcal{O}_E = \mathcal{O}_L$. This seems simply false to me - $\mathcal{O}_E$ represents the third person view, whereas $\mathcal{O}_L$ represents the first person view. You can define $\mathcal{O}_E$ as the set of images representing all possible environment states from the third person perspective and $\mathcal{O}_L$ as the set of images representing all possible environment states from the first person perspective. These are two different sets, but the table seems to be saying they are the same.
> > > > > >
> > > > > > - Row 2: Table 1 says that for [1] "the demonstrations do not include $\mathcal{O}_L$" does not hold - in other words, [1] requires the demonstrations to include $\mathcal{O}_L$. Again, this is simply false. In [1], all the demonstrations are from the third person view (i.e. $\mathcal{O}_E$).
> > > > > >
> > > > > > - Row 3: Table 1 says that for [1], "The learner does not require $\mathcal{O}_E + \mathcal{O}_L$ all the time" does not hold, meaning that "The learner does require $\mathcal{O}_E + \mathcal{O}_L$ all the time" Again this is wrong - in [1], the learner in fact never has to execute the same policy in both observation spaces. [1] in fact makes a weaker assumption than the proposed work does in this regard - it never requires running the same policy in both observation spaces, whereas the present work does.
> > > > > >
> > > > > > - In Row 4: Table 1 says that for [1], "$\mathcal{O}_E$ is not more privileged than $\mathcal{O}_L$ does not hold for [1]. This is not clear - for LBC it is clear that simulator access is more privileged than camera observations, whereas between third and first person views it is not obvious which is more privileged.
> > > > > >
> > > > > > If I am wrong about this, I would really appreciate if the authors could point me to the _exact_ part of [1] which says differently. As it is though, I think this table is very misleading as it suggests that [1] operates under stronger assumptions than the proposed approach, when in fact it seems to me like the opposite it true.

---

> > > > > > > ### Author Response · Authors · 2022-12-10
> > > > > > > **Different viewpoints in [1] are not heterogeneous observations in our work**
> > > > > > >
> > > > > > > Thanks for Reviewer j2UZ’s response. We find that the reviewer's judgment seems to result from the wrong understanding of the heterogeneous observation spaces. The heterogeneous observation spaces denote that the modalities, features, and sensors (types) of two observations can be totally different, in contrast to the **change of “angles of view”** but with the **same type of sensor (camera)** in [1].
> > > > > > >
> > > > > > > Q:
> > > > > > >
> > > > > > > > Table 1 seems wrong to me.
> > > > > > >
> > > > > > > and
> > > > > > >
> > > > > > > > I would really appreciate if the authors could point me to the exact part of [1] which says differently.
> > > > > > >
> > > > > > > A:
> > > > > > >
> > > > > > > For row 1, the feature spaces considered in [1] are the same. Please refer to the caption of the architecture diagram in Figure 2 in [1]:
> > > > > > >
> > > > > > > * " **Images** at time t and t + 4 are sent through a feature extractor to obtain F(ot) and F(ot+4)."
> > > > > > >
> > > > > > > This caption stated that the inputs of [1] are all images. Meanwhile, [1] stated that they dealt with changed camera angles images. For example, in [1]:
> > > > > > >
> > > > > > > * "The color of the target and the **camera angle change** between domains" in Section 6.1;
> > > > > > > * "The **camera angle**, the length of the arms, and the color of the target point are **changed** between domains" in Section 6.1;
> > > > > > > * "we exam the final reward obtained by a policy trained with third person imitation learning vs the **camera angle difference** between the first-person and third-person perspective" in Section 6.2.
> > > > > > >
> > > > > > > Besides, the experiment setups of $\mathcal{O}_E$ and $\mathcal{O}_L$ in our work and [1] are compared as follows,
> > > > > > >
> > > > > > > |                                     |     The features of $\mathcal{O}_E$      |     The features of $\mathcal{O}_L$    |     The dimensions of $\mathcal{O}_E$    |     The dimensions of $\mathcal{O}_L$    |
> > > > > > > |-----------------------------------|----------------------------------------|--------------------------------------|----------------------------------------|----------------------------------------|
> > > > > > > |     Atari experiment in our work    |                   Image                  |        Random Access Memory (RAM)      |          $84 \times 84 \times 4$         |                    $128$                   |
> > > > > > > |       Control experiment in [1]     |                   Image                  |                  Image                 |          $50 \times 50 \times 3$         |          $50 \times 50 \times 3$         |
> > > > > > > |                                     |                                          |                                        |                                          |                                          |
> > > > > > >
> > > > > > > More details of the experimental setup can be found in Section D in the appendix of our paper and Section B in the appendix of [1]. We can see that both the statement and the experiment in [1] supported that the feature spaces considered in [1] are the same.
> > > > > > >
> > > > > > > For rows 2, 3, and 4, the prerequisite to discussing these rows is that the method of [1] is able to solve the setting of row 1. Any method that cannot solve row 1 may still satisfy rows 2-4, such as classical imitation learning algorithms. But obviously, they cannot solve our problem, since allowing "$\mathcal{O}_E \neq \mathcal{O}_L$" is the basic prerequisite to solving our problem. So it is unfair thus incorrect to discuss rows 2-4 for [1].

---

> > > > > > > > ### Comment · Reviewer_j2UZ · 2022-12-12
> > > > > > > > **still not convinced**
> > > > > > > >
> > > > > > > > Sure, I agree that in [1] the inputs for both the expert and the learner are images. But my point is that $\mathcal{O}_E$ and $\mathcal{O}_L$ consist of **disjoint** sets of images. For example, in the Point and Reacher experiments in [1], no matter which actions the learner executes, it will never generate an image in the expert dataset because the targets have different colors and there is no action which changes the color. A similar point holds for viewpoint mismatch. It is therefore inaccurate to say that $\mathcal{O}_E = \mathcal{O}_L$.
> > > > > > > >
> > > > > > > > I think it is misleading to say that $\mathcal{O}_E = \mathcal{O}_L$ simply because they both consist of images, because the two sets of images are disjoint, and the meaning of each pixel is fundamentally different between the two domains. In fact, if the only criteria for saying that $\mathcal{O}_E = \mathcal{O}_L$ is that the inputs have the same shape, you could apply the same argument to vacuously say that _any_ two feature spaces are the same. For example, take your Atari experiments using the RAM states and images. One could construct 84x84x4 "image" representations of the RAM state by setting the first 128 pixels of the first channel to be the RAM values and the rest zeros. Now the input of both domains are images - does this mean that $\mathcal{O}_E = \mathcal{O}_L$? Obviously not, because the _meaning_ of the pixel values in the two domains is fundamentally different. The same holds true in the case of different viewpoints in [1].

---

> > > > > > > > > ### Author Response · Authors · 2022-12-13
> > > > > > > > > **We are glad to see that the agreement of Reviewer j2UZ on the fundamental opinion**
> > > > > > > > >
> > > > > > > > > Thanks for Reviewer j2UZ’s response.
> > > > > > > > >
> > > > > > > > > We are glad to see that the reviewer agreed with the fundamental opinion that [1] considered the **same observation space** (“in [1] the inputs for both the expert and the learner are images”) while we considered **different observation spaces**. The problem settings of [1] and our work are totally different.
> > > > > > > > >
> > > > > > > > > Given the different problem settings, the fundamental challenges are quite different. As illustrated in [1], [1] considered the viewpoint change of the camera, which belongs to the “disjoint domain” under the same observation space, while we aimed to solve the problem of observation space change. They belong to two branches of imitation learning research. As we have emphasized in the previous response (https://openreview.net/forum?id=3ULaIHxn9u7&noteId=_bDZZMD6osj), there have been some other works [2, 3, 4] obtained promising results on the research branch of different observation spaces. Our work takes a step further on this problem. The method in [1] can neither solve their problems nor ours.
> > > > > > > > >
> > > > > > > > > Table 1 in our main paper actually illustrates the above difference. **In Sections 3.1 of the main paper, we have defined $\mathcal{O}_E$ and $\mathcal{O}_L$ are two different observation spaces in contrast to two “disjoint domains” under the same observation space**. It is therefore accurate to say that $\mathcal{O}_E = \mathcal{O}_L$.

---

> > > > > > > > > > ### Comment · Reviewer_N1Ya · 2022-12-14
> > > > > > > > > > **Suggestion: The authors need to make Table 1 clearer to avoid potential misunderstandings**
> > > > > > > > > >
> > > > > > > > > > Thanks for all discussions between Reviewer j2UZ and the authors. The authors’ response is convincing to me, that [1] and this work belong to different learning problems. I also think Table 1 could raise some misunderstandings. As domain adaptation and your problem are different in terms of feature spaces, I think the authors should make it clearer in Table 1: replacing “&#10006;” to “N/A” (not applicable) in the last three lines of the DSIL column to avoid some potential misunderstandings.

---

> > > > > > > > > > > ### Author Response · Authors · 2022-12-14
> > > > > > > > > > > **Thanks for the suggestions**
> > > > > > > > > > >
> > > > > > > > > > > We would like to thank Reviewer N1Ya for the constructive suggestions. We will modify the related part to improve the clarifications. Sincerely we would also like to thank Reviewer j2UZ for the in-depth discussions.

---

### Official Review · Reviewer_w9cE · 2022-10-28

**Confidence:** 4
**Correctness:** 4
**Technical Novelty And Significance:** 3
**Empirical Novelty And Significance:** 3
**Recommendation:** 8

**Clarity, Quality, Novelty And Reproducibility:**

This paper is clearly written, with high-quality explanations and derivations of the approach. The care given to describing the heterogeneous observation setting and how it differs from settings studied in prior work are additionally very helpful. This problem setting and proposed algorithm are both novel.


**Strength And Weaknesses:**

I really like this approach, and the motivation behind the HOIL problem. I think it’s definitely true in many robotics settings that there is a mismatch in observation spaces between the expert and the learner, and addressing this is important.

IWRE as an algorithm is also well-motivated, and follows naturally from a large body of prior work; combining importance weighting and rejection sampling in this setting is natural and novel, and clearly the results speak for themselves.

The one weakness I have of this work is with the evaluation; technically, to prove the viability of the algorithm, the existing “synthetic” observation splits in the Atari and Mujoco environments make sense, but I’d really love to see a more realistic evaluation where heterogeneous observations are actually ecologically viable — perhaps in settings in robotic manipulation for example.


**Summary Of The Paper:**

The bulk of existing work on imitation learning focuses on settings in which the expert demonstrator and learner operate under the **same observation space**; however, in many real-world settings, this isn’t realistic. The work formalizes the problem of heterogeneously observable imitation learning (HOIL) and present an algorithm — Importance Weighting with Rejection (IWRE) to address this.

Concretely, given a set of expert demonstrations in a given observation space, the HOIL setting this work looks at decomposes learning into collecting a set of (suboptimal) “matching correspondence” data online, running a policy that allows one to correlate expert observations with learner observations (even when there’s no explicit overlap in observables — e.g., Atari ALE RAM state vs. visual features). The proposed IWRE framework then tries to address two key problems in this learning setup via an online IRL-like procedure (similar in nature to GAIL); first, the problem of optimality/dynamics mismatch — the “expert demonstrations” are assumed to be optimal, while the correspondence data is not; as a result, directly imitating from the union of both datasets would result in a bad policy. To get around this, similar to prior work in offline RL, this work proposes using importance sampling (the IW in IWRE) to correct for this mismatch, prioritizing the expert demonstrations.

The second problem IWRE addresses is that of a support mismatch; the “correspondence matching” data is again problematic because its suboptimality might mean that the corresponding data only sees a fraction of the state space. As a result, IWRE uses rejection sampling to drive the learner’s behavior policy towards the support of the actual expert observation distribution (exploiting a learned discriminator that partitions the expert demos and correspondence demos), therefore circumventing this issue.

The IWRE approach is evaluated on a set of (albeit somewhat synthetic) environments from the Atari Arcade Learning Environment (expert observations — frames, learner observations — RAM state), and Mujoco Continuous control (expert/learner see disjoint halves of the Mujoco observation space). Compared to traditional imitation learning baselines and reinforcement learning upper bounds (for ceiling results), the IWRE approach is strong, performant, and seems to generalize across environments.


**Summary Of The Review:**

I believe this to be strong work, with an evaluation that demonstrates the viability of the proposed approach and difficulty of the problem setting. However, I’d love to see more ecologically viable/realistic evaluations, rather than the somewhat synthetic tasks studied in this work.

---

> ### Author Response · Authors · 2022-11-15
> **Authors Response**
>
> We thank Reviewer w9cE for the comments and suggestions. Below we would like to respond to the evaluation part of our work.
>
> **Q1**:
>  >to prove the viability of the algorithm, the existing “synthetic” observation splits in the Atari and Mujoco environments make sense, but I’d really love to see a more realistic evaluation where heterogeneous observations are actually ecologically viable.
>
> **A1**:
>
> We are glad to hear the reviewer's positive comments about the evaluation viability of our experiments. Indeed, the Mujoco environments are synthetic to form the heterogeneous observations as the HOIL problems. On the other hand, the heterogeneous observations (pixel observations and RAM observations) exist naturally in Atari domains, while few works took them into consideration. Meanwhile, we believe that our approach can be widely applied in real-world HOIL scenarios, such as AI diagnosis, autonomous driving, and robotic manipulation. We treat them as important future works.

---

### Official Review · Reviewer_N1Ya · 2022-12-02

**Confidence:** 5
**Correctness:** 4
**Technical Novelty And Significance:** 4
**Empirical Novelty And Significance:** 3
**Recommendation:** 10

**Clarity, Quality, Novelty And Reproducibility:**

The writing of this paper is clear and easy to follow. The related works are well discussed, and the experiment is sufficient. The combination of importance weighting and rejection learning is quite novel in the reinforcement learning and imitation learning communities. The paper contained enough details for reproducibility.

**Strength And Weaknesses:**

Strengths：
* This paper studied a challenging but more general learning scenario, with weaker assumptions than related works. Also, the HOIL problem is well-defined.

* For the algorithm part, the analysis of the dynamics mismatch and the support mismatch is quite convincing, with clear mathematical formulations and visualization empirical studies. Meanwhile, the motivated IWRE makes sense.

Weaknesses：

The notations of this work are somehow heavy in this paper. It would be better to simplify some notations.

**Summary Of The Paper:**

This paper aims to solve a branch of imitation learning scenario that the expert’s and the learner’s feature (observation) spaces are different. This scenario is quite difficult compared with existing imitation learning work under the same observation space. Moreover, the related works relied on the expert’s and learner’s observations can be obtained during the whole learning process, which can be costly in many real-world applications. So this paper tried to solve this problem with limited observations coexistence.

At first, this paper formulated the learning problem as heterogeneously observable imitation learning (HOIL), where some expert demonstrations and an auxiliary policy were given under the expert’s observation space, and the target was to learn a policy under the learner’s observation space. Since these two spaces can be arbitrarily different, they needed the auxiliary policy to sample some data containing both two observations as the bridge between the expert demonstrations and the learned policy. Later on, they limited the number of observation coexistence to lower the cost.

By analyzing this problem, they found the underlying challenges, i.e., the dynamics mismatch and the support mismatch. These two mismatches appeared due to the imperfection of the auxiliary policy. To solve this problem, they proposed an algorithm of importance weighting and rejection (IWRE), where the IW solved the dynamics mismatch and the RE solved the support mismatch.

To evaluate the performance of IWRE, they did some experiments on MuJoCo and Atari benchmarks. Compared with the ablation, domain-adaptation imitation learning methods, and other heterogeneous observation methods, IWRE obtained the best performance among these environments.

**Summary Of The Review:**

This paper studied a challenging but more realistic imitation learning scenario, in which the expert’s and the learner’s observations are heterogeneous. Also, the authors proposed a novel and reasonable method IWRE to solve this problem. The contributions of this work are sufficient. So I recommend this paper be accepted.

After rebuttal:

I have gone through the responses and discussions between the co-reviewers and the authors. I think the authors have clearly answered the questions of other reviewers, and thus believe this paper is a good submission. So I increased my score and stand for accepting it.

---

> ### Author Response · Authors · 2022-12-02
> **Authors Response**
>
> We thank Reviewer N1Ya for the comments and suggestions. Below we would like to discuss the notations of our work.
>
> > The notations of this work are somehow heavy in this paper. It would be better to simplify some notations.
>
> Compared to imitation learning problems under one observation space, our setting has two spaces, so more detailed notations are necessary. We have gathered all notations in Table 1 in the supplementary material. Thanks for your suggestion. We will further improve the notations.

---

### Decision · Program_Chairs · 2023-01-20

**Decision:**

Accept: notable-top-25%

**Justification For Why Not Higher Score:**

While interesting in its setting and technical approach, the method's evaluation falls short of demonstrating the usefulness of the approach in practical settings.

**Justification For Why Not Lower Score:**

The paper addresses an important open problem in learning from demonstrations with an intuitive and technically interesting approach.

**Metareview: Summary, Strengths And Weaknesses:**

Strengths:
- Addresses an important open problem in learning from demonstrations of learning when demos don't share the agent's observation space. Having only a small "correspondence" dataset that has access to both observation spaces also makes the algorithm more broadly applicable than approaches such as "learning by cheating".
- Well-motivated algorithm design with a technically interesting combination of importance weighting and rejection sampling. Seems straightforward to implement.
- Strong empirical results on their chosen evaluation settings.

Weakness:
- Table 1 contains a somewhat misleading representation of prior works that dealt with different viewpoints of a scene as having the "same observation space", and the author discussion with reviewer j2UZ appears to focus mainly on the semantics of whether the sensors are the same or different. While the original review from j2UZ contained inaccuracies, I agree with them on the points argued in the discussion phase that this is not really a meaningful distinction from having "different observation spaces" --- top and frontal camera views of a scene are just as much different "observation spaces" as having proprioceptive and exteroceptive views of the same scene. I would urge the authors to improve the writing around these claims.
     - While prior work such as third-person imitation learning (Stadie et al 2017) has addressed related settings, they rely on the existence of some relationships between the views, which allows for classes of methods (domain adaptation) that do not apply to the more dramatically different "observation spaces" considered in this work, for which some kind of correspondence dataset seems inevitable, and this algorithmic choice seems justified.

- The evaluation focuses on rather simplistic and unrealistic settings, namely RAM state versus visual observations in Atari, and disjoint halves of the state space in MuJoCo. This detracts significantly from our ability to evaluate whether the proposed approach is in fact useful in practical settings.

**Note From Pc:**

if the above contains the word "oral" or "spotlight" please see: "oral" presentation means -> notable-top-5% and "spotlight" means -> notable-top-25%. As stated in our emails, we are disassociating presentation type from AC recommendations